# FedorAS: Federated Architecture Search under system heterogeneity

## Abstract

Federated learning (FL) has recently gained considerable attention due to its ability to learn on decentralised data while preserving client privacy. However, it also poses additional challenges related to the heterogeneity of the participating devices, both in terms of their computational capabilities and contributed data. Meanwhile, Neural Architecture Search (NAS) has been successfully used with centralised datasets, producing state-of-the-art results in constrained or unconstrained settings. However, such centralised datasets may not be always available for training. Most recent work at the intersection of NAS and FL attempts to alleviate this issue in a cross-silo federated setting, which assumes homogeneous compute environments with datacenter-grade hardware. In this paper we explore the question of whether we can design architectures of different footprints in a cross-device federated setting, where the device landscape, availability and scale are very different. To this end, we design our system, `FedorAS`, to discover and train promising architectures in a resource-aware manner when dealing with devices of varying capabilities holding non-IID distributed data. We present empirical evidence of its effectiveness across different settings, spanning across three different modalities (vision, speech, text), and showcase its better performance compared to state-of-the-art federated solutions, while maintaining resource efficiency.

## 1 Introduction

As smart devices become omnipresent where we live, work and socialise, the ML-powered services that these provide grow in sophistication. This ambient intelligence has undoubtedly been sustained by recent advances in Deep Learning (DL) across a multitude of tasks and modalities. Parallel to this race for state-of-the-art performance in various in DL benchmarks, mobile and embedded devices also became more capable to accommodate new Deep Neural Network (DNN) designs [37], some even integrating specialised accelerators to their System-On-Chips (SoC) (e.g. NPUs) to efficiently run DL workloads [3]. These often come in various configurations in terms of their compute/memory capabilities and power envelopes [4] and co-exist in the wild as a rich multi-generational ecosystem (*system heterogeneity*) [79]. These devices bring intelligence through users' interactions, also innately heterogeneous amongst them, leading to non-independent or identically distributed (non-IID) data in the wild (*data heterogeneity*).

Powered by the recent advances in SoCs' capabilities and motivated by privacy concerns [74] over the custody of data, Federated Learning (FL) [58] has emerged as a way of training on-device on user data without it ever directly leaving the device premises. However, FL training has largely been focused on the weights of a static global model architecture, assumed to be runnable by every participating client [40]. Not only may this not be the case, but it can also lead to subpar performance of the overall training process in the presence of stragglers or biases in the case of consistently dropping certain low-powered devices. On the opposite end, more capable devices might not fully take advantage of their data if the deployed model is of reduced capacity to ensure all devices can participate [52].

Parallel to these trends, Neural Architecture Search (NAS) has become the *de facto* mechanism to automate the design of DNNs that can meet the requirements (e.g. latency, model size) for these to run on resource-constrained devices. The success of NAS can be partly attributed to the fact that these frameworks are commonly run in datacenters, where high-performing hardware and/or large curated

datasets [43, 23, 20, 39, 62, 62] are available. However, this also imposes two major limitations on current NAS approaches: i) *privacy*, i.e. these methods were often not designed to work in situations when user's data must remain on-device; and, consequently, ii) *tail data non-discoverability*, i.e. they might never be exposed to infrequent or time/user-specific data that exist in the wild but not necessarily in centralized datasets. On top of these, the whole cost is born by the provider and separate on-device modelling/profiling needs to be done in the case of hardware-aware NAS [26, 73, 45], which has mainly focused on inference performance hitherto.

Motivated by the aforementioned phenomena and limitations of the existing NAS methods, we propose `FedorAS`, a system that performs NAS over *heterogeneous devices* holding *heterogeneous data* in a resource-aware and federated manner. To this direction, we cluster clients into tiers based on their capabilities and design supernet comprising operations covering the whole spectrum of compute complexities. This supernet acts both as search space and a weight-sharing backbone. Upon federation, it is only partially and stochastically shared to clients, respecting their computational and bandwidth capabilities. In turn, we leverage resource-aware one-shot path sampling [28] and adapt it to facilitate lightweight on-device NAS. In this way, networks in a given search space are not only deployed in a resource-aware manner, but also trained as such, by tuning the downstream communication (i.e. the subspace explored by each client) and computation (i.e. FLOPs of sampled paths) to meet the device's training budget. Once federated training of the supernetwork has completed, usable pretrained networks can be extracted even before performing fine-tuning or personalising per device, thus minimising the number of extra on-device training rounds to achieve competitive performance.

In summary, in this work we make the following contributions:

- We propose a system for resource efficient federated NAS that can be applied in *cross-device* settings, where partial device participation, device and data heterogeneity are innate characteristics.
- We implement a system called `FedorAS` (Federated nAS) that leverages a server-resident supernet enabling weight sharing for efficient knowledge exchange between clients, without directly sharing common model architectures with one another.
- We propose a novel aggregation method named OPA (OPerator Aggregation) for weighing updates from multiple "single-path one-shot" client updates in a frequency-aware manner.
- We empirically evaluate the performance and convergence of our system under IID and non-IID settings across different datasets, tasks and modalities, spanning different device distributions and compare our system's behaviour against state-of-the-art FL techniques.

## 2 BACKGROUND & MOTIVATION

**Federated Learning.** A typical FL pipeline is comprised of three distinct stages: given a *global* model initialised on a central server, $\omega_g^{(t=0)}$, *i)* the server randomly samples $k$ clients out of the available $K$ ($k \ll K$ for *cross-device*; [10] $k = K$ for *cross-silo* setting) and sends them the current state of the global model; *ii)* those $k$ clients perform training on-device using their own data partition, $D_i$, for a number of epochs and send the updated models, $\omega_i^{(t)}$, back to the server after local training is completed; finally, *iii)* the server aggregates these models and a new *global* model, $\omega_g^{(t+1)}$, is obtained. This aggregation can be implemented in different ways [58, 70, 53]. For example, in FedAvg [58] each update is weighted by the relative amount of data on each client: $\omega_g^{(t+1)} = \sum_{i=0}^{k} \frac{|D_i|}{\sum_{j=0}^{k} |D_j|} \omega_i^{(t)}$.

Stages *ii)* and *iii)* repeat until convergence. The quality of the *global* model $\omega_g$ can be assessed: on the *global* test set; by evaluating the fit of the $\omega_g$ to each participating client's data ($D_i$) and derive fairness metrics [54]; or, by evaluating the adaptability of the $\omega_g$ to each client's data or new data these might generate over time, this is commonly referred to as *personalised FL* [27, 51].

Contrary to traditional distributed learning, cross-device FL performs the bulk of the compute on a highly heterogeneous [40] set of devices in terms of their compute capabilities, availability and data distribution. In such scenarios, a trade-off between model capacity and client participation arises: larger architectures might result in more accurate models which may only be trained on a fraction of the available devices; on the other hand, deploying smaller footprint networks could target more devices – and thus more data – for training, but these might be of inferior quality (gap in Fig. 1).

**Neural Architecture Search.** NAS is usually defined as a bi-level optimisation problem:

$$a^\star = \arg\min_{a \in \mathbb{A}} \mathcal{L}(\omega_a^*(D_t), D_v), \quad \text{where} \quad \omega_a^*(D_t) = \arg\min_{\omega_a} \mathcal{L}(\omega_a, D_t) \tag{1}$$

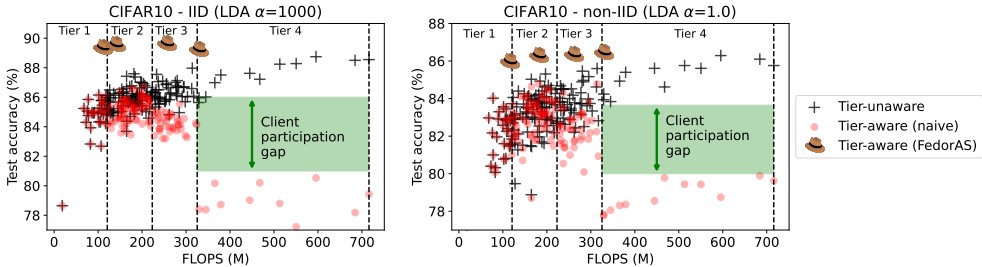

**Figure 1:** For three LDA settings, 160 architectures are randomly sampled from a ResNet-like search space and trained on a CIFAR-10 FL setup with 100 clients, with 10 clients participating on each round for a total of 500 rounds. Clients are uniformly assigned to a tier, resulting in 25 clients per tier. Given sufficient data and ignoring tier limits, model performance tends to improve as its footprint (FLOPS) increases (black crosses). However, when models are restricted to only train on clients that support them (tier-aware), the lack of data severely restricts the performance of more capable models (red dots). `FedorAS` successfully overcomes the challenges of tier-aware FL and outperforms existing system heterogeneous baselines, as shown later in Fig. 3.

where $\mathbb{A}$ is a finite (discrete) set of architectures to search from (a *search space*), $\mathcal{L}$ is a loss function, $\omega_a$ are weights of a model with architecture $a$, and $D_{\{v,t\}}$ are validation and training datasets, respectively. The main challenge of NAS comes directly from the fact that in order to assess quality of different architectures (*outer optimisation*), we have to obtain their optimal weights which entitles conducting full training (*inner optimisation*).

There exist multiple approaches to speed up NAS [65, 56, 25, 16, 26, 87, 1, 57, 71, 28, 59]. More relevant to our work are those utilising the concept of a supernet [15, 7]; where a single model that encapsulates all possible architectures from the search space is created and trained. Specifically, a supernet is constructed by defining an operator that incorporates all candidate operations (the set of which we denote by $\mathbb{O}_l$) for each searchable layer $l$. A common choice is to define it as a weighted sum of candidates' individual outputs $y_l = \sum_{o \in \mathbb{O}_l} \alpha_l^{(o)} * o(y_{l-1})$, where factors $\{\alpha_l^{(o)}\}_{o \in \mathbb{O}_l}$ of each layer $l$ can be defined differently for different methods (e.g, continuous parameters [56, 16, 25] or random one-hot vectors [28]). Importantly for us, methods that use sparse weighting factors can avoid executing operations associated with zero weights, saving both memory and compute [16, 28].

After a supernet has been constructed and trained, searching for $a^\star$ is usually performed by either investigating architectural parameters [56, 25, 16], or using zero-th order optimisation methods to directly solve the outer loop of Eq. 1 while approximating $\omega_a^*$ with weights taken from the supernet (thus avoiding the costly inner optimisation) [48, 28]. The final model can then be retrained in isolation using either random initialisation or weights from the supernet as a starting point.

**Challenges of Federated NAS.** As highlighted before, devices in the wild exhibit different compute capabilities and can hold non-IID distributed data, resulting in system and data heterogeneity. In the context of NAS, system heterogeneity has a particularly significant effects, as we might no longer be able to guarantee that any model from our search space can be efficiently trained on all devices. This inability can be attributed to insufficient compute power, limited network bandwidth or unavailability of the client at hand. Consequently, some of the models might be deemed worse than others not because of their worse ability to generalise, but because they might not be exposed to the same subsets of data as others – as shown in Fig. 1, where we show models of different footprint trained across client of varying capabilities under constrained (*tier-aware*) and full (*tier-unaware*) participation.

## 3 THE FEDORAS FRAMEWORK

`FedorAS` is a resource-aware Federated NAS framework that combines: learning from clients across all tiers and yielding models tailored to each tier that benefit from this collective knowledge.

**Workflow.** `FedorAS`' workflow consists of three stages (Fig. 2): *i) supernet training*, *ii) model search and validation* and *iii) model fine-tuning*. First, we train the supernet in a resource-aware and federated manner (**Stage 1**, Sec. 3.1). We then search for models from the supernet with the goal of finding the best architecture per tier (**Stage 2**, Sec. 3.2). Models are effectively sampled, validated on a global validation set and ranked per tier. These architectures and their associated weights act as initialisation to the next phase, where each model is fine-tuned in a per-tier manner (**Stage 3**, Sec. 3.3). The end goal of our system is to have the best possible model per each cluster of devices.

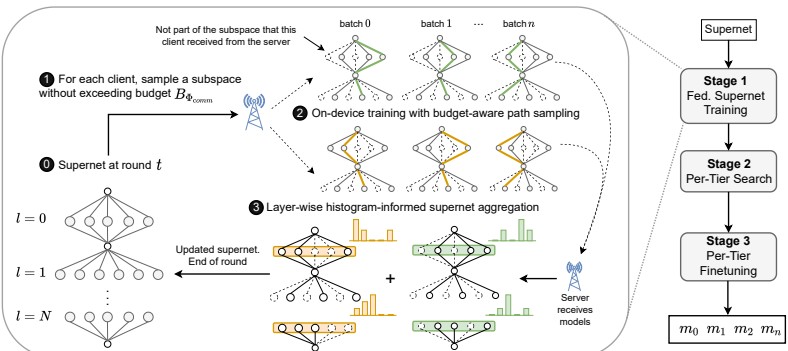

**Figure 2:** Training process workflow with `FedorAS`.

**Design rationale.** We build our system around the concept of a supernet to facilitate weight-sharing between architectures of various footprints. Operations in the supernet are samplable from paths (i.e. models) of different footprint. As such, while normally large models would not be directly trained on data of low-tier clients, our design allows for indirectly sharing this knowledge through the association of the same operation to different paths. To ensure efficient training of a supernet, we chose to base our approach on SPOS [28] and adapt it (see Eq. 4) since its training procedure is lightweight, introducing very little overhead in terms of memory consumption and/or required floating-point operations, especially compared to [56]. Last, we opted for clustering devices into "tiers" based on their computational capabilities and search for an architecture for each tier as a balance between having one model to fit all needs [30] and a different architecture per client [60].

## 3.1 SUPERNET TRAINING

**❶ Search space & models.** First, we define the search space in terms of a range of operators options per layer in the network that are valid choices to form a network. This search space resides as a whole on the server and is only partially communicated to participating clients of a training round to keep communication overheads minimal. Specific models (i.e. paths) consist of selections of one operator option per layer, sampled in a single-path-one-shot manner on-device per local iteration.

**❶ Subspace sampling.** It is obvious that communicating the whole space of operators along with the associated weights to each device becomes quickly intractable, especially bearing in mind that communication is usually a primary bottleneck in Federated Learning [40, 52]. To this direction, `FedorAS` adopts a uniform parameter size budget, $B_{\Phi_{\text{comm}}}$, and samples[1] the search space for operators until this limit is hit (Eq. 2). Setting the limit to half the size of a typical network deployed for a task worked sufficiently well in our experiments and in fact accelerated convergence (Sec. 4.4).

$$\sum_{l=0}^{L} \sum_{o \in \mathbb{O}_l} \mathbb{1}_{\hat{\mathbb{O}}_l}(o) \Phi_{\text{comm}}(o) < B_{\Phi_{\text{comm}}}, \hat{\mathbb{O}}_l \subseteq \mathbb{O}_l \tag{2}$$

where $L$ is the number of layer in the supernet, $\mathbb{O}_l$ the candidate and $\hat{\mathbb{O}}_l$ the selected operations in layer $l$, $\mathbb{1}$ a unit vector of valid operations and $\Phi_{\text{comm}}$ a measure of DNN size (e.g. #parameters).

In terms of sampling strategies, we experimented with uniform operator sampling, which we found to work sufficiently well and provide uniform coverage over the search space. It is also worth noting that a different subspace (not necessarily mutex) could be selected for each participating client in a round.

**❷ Client-side sampling & local training.** Participating clients receive the subspace sampled on the server, $\{\hat{\mathbb{O}}_l\}_{l=0}^{L}$, from which they sample a single operator on every layer. This constitutes a path along the supernet ($p_L$) representing a single model. For every batch, clients sample paths that do not surpass the assigned training budget $B_{\Phi_{\text{train}}}^{\text{Tier}}$. Throughout this work, we consider $\Phi_{\text{train}}(\cdot)$ to be a cost function that counts the FLOPs of a given operator. This FLOPs limit is defined per tier so that a network does not exceed the capabilities of the target device. Our goal is to sample valid paths uniformly, to ensure systematic coverage of the entire (sub) search space during training:

$$p_L = \bigcup_{l=0}^{L} o_l \sim \mathcal{U}\{\hat{\mathbb{O}}_l\} \text{ s.t. } \sum_{l=0}^{L} \Phi_{\text{train}}(o_l) < B_{\Phi_{\text{train}}}^{\text{Tier}} \tag{3}$$

---

[1]Non-parametric operations are always sent downstream and layers without such options are prioritised to guarantee a valid network.

However, realising Eq. 3 efficiently is not a trivial task. Originally [28], the authors considered a naive approach of repeatedly sampling a path until it fits the given budget, which results in non-negligible overhead if the probability of finding a model under the threshold is low. Were we to employ such a method, the most restricted devices, for which the set of eligible models is the smallest, would be the ones burdened with the largest overhead. Therefore, we propose a greedy approximation in which operations are selected sequentially. Specifically, in order to obtain a path $p_L = \{o_i\}_{i=0}^{L}$ we sample operations layer-by-layer, according to a random permutation $\sigma$, in such a way that the $i$-th operation is chosen from the candidates for layer $\sigma(i)$ whose total overhead would not violate the constraint:

$$o_i \sim \mathcal{U}\{\, o : o \in \hat{\mathbb{O}}_{\sigma(i)} \wedge \sum_{j=0}^{i} \Phi_{\text{train}}(o_j) + \Phi_{\text{train}}(o) < B_{\Phi_{\text{train}}}^{\text{Tier}} \,\} \tag{4}$$

We can ensure that Eq. 4 can always obtain a valid architecture without resampling if layers have Identity among their candidate operations and prioritising the selection of those which do not.

After having sampled the path, a model is instantiated and a single optimization step using a single batch of data is performed. The number of samples passing through each operator are kept and communicated back to the server, along with the updates, for properly weighting updated parameters upon aggregation, as we will see next.

❸ **Aggregation with OPA.** An operator gets stochastically exposed to clients data. This stems from *subspace sampling* and client-side *path sampling*. As such, naively aggregating updates in an equi-weighed manner (Plain-Avg) or total samples residing on a client (FedAvg [58]) would not reflect the training experience during a round. For this reason, we propose OPA, OPerator Aggregation, an aggregation method that weights updates based on the relative experience of an operator across all clients that have updated that operator. Concretely, our method is generalisation of FedAvg where normalisation is performed independently for each layer, rather than collectively for full models. In order to enable that, we keep track of how many examples were used to update each searchable operation $o_l$, independently on each client, and later use this information to weight updates. Formally:

$$\omega_g^{(t+1)}(o_l) = \begin{cases} \sum_{i=0}^{k} \frac{|D_{i,o_l}^{(t)}|}{\sum_{j=0}^{k} |D_{j,o_l}^{(t)}|} \omega_i^{(t)}(o_l) & \text{if } |C_{o_l}^{(t)}| > 1 \\ \omega_g^{(t)}(o_l) & \text{otherwise} \end{cases} \tag{5}$$

where $\omega_g^{(t)}(\cdot)$ are global weights, $\omega_i^{(t)}(\cdot)$ are local weights of client $i$ at global step $t$, $|D_{i,o_l}^{(t)}|$ is the number of samples having backpropagated through an operator $o_l$ for client $i$ in round $t$, and $C_{o_l}^{(t)}$ is the set of clients $i$ s.t. $|D_{i,o_l}^{(t)}| > 0$. Updates to $\omega_g^{(t)}$ happen only if $|C_{o_l}^{(t)}| > 1$ in order to protect the privacy of single clients. Finally, if an operation is always selected, then Eq. 5 recovers FedAvg, which means we can effectively use OPA throughout the model and not only for searchable layers.

## 3.2 MODEL SEARCH & VALIDATION

After training the supernet, a search phase is implemented to discover the best trained architecture per device tier. Models are sampled with NSGA-II [22] and evaluated on a global validation set. The rationale behind selecting the NSGA-II algorithm for our search is the fact that it is multi-objective and allows us for efficient space exploration with the goal of optimising for model size in a specific tier and accuracy. Other search algorithms can be used in place of NSGA-II, based on the objective at hand. Search can stop after a specified number of steps or when an accuracy threshold is met. At the end of this stage, we have a model architecture per tier, already encompassing knowledge across devices of different tiers, which serves as initialisation for further training.

## 3.3 FINE-TUNING PHASE

Subsequently, we move to train each of the previously produced models in a federated manner across all eligible clients. This means that architectures falling in a specific tier, in terms of footprint, can be trained across clients belonging to that tier and above. Our hypothesis here is that compared to regular FL where one needs to trade off capacity of participation, `FedorAS` allows for better generalisation due to knowledge sharing through its supernet structure. Here we use conventional FedAvg and partial client participation per round, again with the goal of training a single network per tier.

**Table 1:** Datasets for evaluating `FedorAS`. We partition CIFAR-10/100 following the Latent Dirichlet Allocation (LDA) partitioning [36], with each client receiving approximately equisized training sets. For CIFAR-10, we consider $\alpha \in \{1000, 1.0, 0.1\}$ configurations, while for CIFAR-100, we adopt $\alpha = 0.1$ [70]. The remaining datasets come naturally partitioned.

| Dataset | Search Space | Number Clients | Number Examples | Target Task |
|---|---|---|---|---|
| CIFAR10 | CNN-L | 100 | $50K$ | Image classification |
| CIFAR100 | CNN-L | 500 | $50K$ | Image classification |
| Speech Commands | CNN-S | $2,112$ | $105.8K$ | Keyword Spotting |
| Shakespeare | RNN | 715 | $38K$ | Next char. prediction |

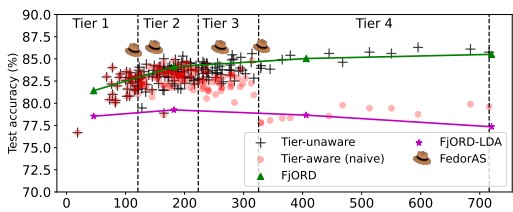

**Figure 3:** `FedorAS` outperforms other approaches (details in Appendix D.7). CIFAR-10 (non-IID, $\alpha$=1.0). FjORD is represented as a line as it can switch between operating points on-the-fly via Ordered Dropout.

## 4 EVALUATION

This section provides a thorough evaluation of `FedorAS` across different tasks to show its performance and generality. First, we compare `FedorAS` to existing approaches in the context of federated NAS in the *cross-device* and *cross silo* setting. Next, we we draw from the broader FL literature and showcase our technique's performance gains compared to *homogeneous* and *heterogeneous* federated solutions (Sec. 4.2). This can be traced back to the benefits of supernet *weight sharing*; as such we, subsequently, quantify its contribution by comparing it to `FedorAS` with randomly initialised networks trained on eligible clients in a federated way (Sec. 4.3) without weight sharing. Last, we showcase the contribution of specific components of our system through an *ablation study* (Sec. 4.4) and also the performance and behaviour of *alternative search methods* (Sec. 4.5).

### 4.1 EXPERIMENTAL SETUP

**Datasets & Baselines.** Datasets are summarised in Tab. 1. More information in Appendix D.

**Search space.** The adopted search spaces are specific to the dataset and task at hand, both in terms of size and type of operators. In general, we assume our highest tier of model sizes to coincide with a typical network footprint used for that task in the FL literature (e.g. ResNet-18 for CIFAR-10). Nevertheless, there may be operators in the space that cannot be sampled in some tiers due to their footprint. `FedorAS` sets the communication budget $B_{\Phi_{comm}}$ to be half the size of the supernet. The full set of available operators per task is provided in the Appendix D.3.

**Clusters definition.** In our experiments, we cluster clients based on the amount of operations they can sustain per round of training (#FLOPs), for simplicity. Other resources (e.g. #parameters, energy consumption) can be used in place or in conjunction with FLOPs. More details in Appendix D.4.

### 4.2 PERFORMANCE EVALUATION

**Federated NAS.** We start the evaluation by comparing our system with existing works in the federated NAS domain. Specifically, we find two systems that perform federated NAS, namely FedNAS [30] and SPIDER [60] and compare in the *cross-device* and *cross-silo* settings. In the former setting, we adapt FedNAS to support partial participation and compare their technique to `FedorAS` under the same *cross-device* settings in CIFAR-10. Results are shown in Tab. 2, showing `FedorAS` performing 1.04% better than FedNAS on CIFAR-10$_{\alpha=1}$ and 48.7% on CIFAR-10$_{\alpha=0.1}$, for the same training memory footprint. Further details can be found in Appendix D.6. At the same time, while running in *cross-silo* setting is not the main focus of `FedorAS`, we have adapted our experimental setting to match that of FedNAS and SPIDER. Results are shown in Tab. 3 with `FedorAS` performing +11.6% and -1.3% than the baselines, respectively, on the test set of their selected settings.

**Table 2:** Cross-device federated NAS on CIFAR-10.

| Dataset | Method | Mem. Peak (MB) | Perf. (%) |
|---|---|---|---|
| CIFAR10$_{\alpha=1}$ | FedNAS | 3837 | **90.02** |
| | FedNAS (adj. batch size) | **1919** | 85.45 |
| | FedorAS | 1996 | 86.46$_{\pm0.32}$ |
| CIFAR10$_{\alpha=0.1}$ | FedNAS | 3837 | 65.28 |
| | FedNAS (adj. batch size) | **1919** | 54.84 |
| | FedorAS | 1996 | **81.53**$_{\pm0.29}$ |

**Table 3:** Cross-silo federated NAS on CIFAR-10.

| Dataset | Method | Validation acc. | Test acc. | #clients |
|---|---|---|---|---|
| CIFAR10$_{\alpha=0.5}$ | FedNAS personalised* | 90.4$_{\pm2.4}$ | - | 20 |
| | FedNAS global | - | 81.2 | 16 |
| | FedorAS cross-silo | **97.2**$_{\pm0.4}$ | **90.6**$_{\pm0.2}$ | 20 |
| CIFAR10$_{\alpha=0.2}$ | SPIDER | - | **92.00**$_{\pm2.0}$ | 8 |
| | FedorAS cross-silo | - | 90.82 | 8 |

*FedNAS reports validation acc for this setting.

**Comparison with Random Search.** We further compare our search with a random search baseline – which is a naive way of running FL NAS in our experimental setting (the same search space, etc.) –

**Table 4:** Comparison with heterogeneous federated baselines. `FedorAS` performs better across datasets.

| Dataset | Method | MFLOPs | Params (M) | Performance |
|---|---|---|---|---|
| CIFAR10$_{\alpha=1000}$ | ZeroFL$_{s=90\%}$ [66] | 557‡ | 11.17 | 82.82$_{\pm0.64}$ |
| | FjORD$_{LDA}$ [34] | [**35**, **139**, 313, 556] | [**0.70**, **2.79**, 6.28, 11.16] | [78.19$_{\pm1.20}$, 78.63$_{\pm1.31}$, 78.25$_{\pm1.06}$, 77.19$_{\pm0.85}$] |
| | `FedorAS`$_{per\ tier}$ | [111, 140, **256**, **329**] | [2.96, 2.93, **3.35**, **4.32**] | [**89.40**$_{\pm0.19}$, **89.60**$_{\pm0.15}$, **89.64**$_{\pm0.22}$, **89.24**$_{\pm0.29}$] |
| CIFAR10$_{\alpha=1}$ | ZeroFL$_{s=90\%}$ [66] | 557‡ | 11.17 | 81.04$_{\pm0.28}$ |
| | FjORD$_{LDA}$ [34] | [**35**, **139**, 313, 556] | [**0.70**, **2.79**, 6.28, 11.16] | [78.54$_{\pm0.12}$, 79.25$_{\pm0.51}$, 78.66$_{\pm0.29}$, 77.35$_{\pm0.44}$] |
| | `FedorAS`$_{per\ tier}$ | [116, 183, **262**, **330**] | [2.59, 2.90, **3.55**, **4.31**] | [**85.99**$_{\pm0.13}$, **86.30**$_{\pm0.41}$, **86.34**$_{\pm0.19}$, **86.46**$_{\pm0.32}$] |
| CIFAR10$_{\alpha=0.1}$ (Acc. (%) ↑ is better) | FjORD$_{LDA}$ [34] | [**35**, **139**, 313, 556] | [**0.70**, **2.79**, 6.28, 11.16] | [61.43$_{\pm0.39}$, 60.81$_{\pm1.42}$, 59.72$_{\pm5.19}$, 57.44$_{\pm3.53}$] |
| | `FedorAS`$_{per\ tier}$ | [117, 159, **238**, **345**] | [2.17, 3.13, **2.49**, **2.61**] | [**81.01**$_{\pm0.46}$, **81.53**$_{\pm0.29}$, **80.64**$_{\pm0.66}$, **80.85**$_{\pm0.28}$] |
| Shakespeare (Perplexity ↓ is better) | FjORD [34] | [**1, 3, 7, 11, 17**] | [**0.01, 0.04, 0.08, 0.14, 0.21**] | [4.44$_{\pm0.07}$, 3.91$_{\pm0.10}$, 3.87$_{\pm0.13}$, 3.87$_{\pm0.13}$, 3.87$_{\pm0.13}$] |
| | `FedorAS`$_{per\ tier}$ | [7, 12, 15, 21, 24] | [0.09, 0.15, 0.18, 0.26, 0.30] | [**3.43**$_{\pm0.01}$, **3.39**$_{\pm0.04}$, **3.38**$_{\pm0.03}$, **3.40**$_{\pm0.01}$, **3.42**$_{\pm0.01}$] |
| SpeechCommands (35 classes) (Accuracy (%) ↑ is better) | Oort [44]† | 2382 | 21.29 | 62.20 |
| | PyramidFL [49]† | 2382 | 21.29 | 63.84 |
| | `FedorAS`$_{best}$★ | **10** | **0.63** | **70.10** |

† [44, 49] perform client selection based on system heterogeneity and are provided for context. FLOPs computed assuming the common [86] 40×51 MFCC features input.
‡ [66] speeds-up training w/ highly-sparse convs, attainable only with specialised h/w. ★ result obtained from the best model of setup in Appendix F.

and find that `FedorAS` performs significantly better at an average of +5.11pp, +3.24pp, +12.84pp, +9.27pp, +0.25p across tiers for CIFAR-10$_{\alpha=\{1000,1,0.1\}}$, CIFAR-100 and Shakespeare, respectively. Only in the case of SpeechCommands did our search result in -0.87pp of accuracy on average. We suspect this is an artifact of insufficiently-long fine-tuning which means models might not have converged (suggested by a high variance). Detailed results are provided in Appendix E.1.

**Federated Learning baselines.** Next, we compare the performance of `FedorAS` with different federated baselines, including *homogeneous* [70] and system *heterogeneous* frameworks [34, 44, 66]. In the former setting, we compare with results from [70] on CIFAR100$_{\alpha=0.1}$. `FedorAS` performs 1 pp[2] better than the FedAvg baseline, at 45.84%, but at a fraction of the cost[3]. Simultaneously, retraining the discovered model from scratch using random initialisation under the same training setting as the baseline results in 11.43 pp higher accuracy than the best FedAdam (63.94% vs 52.50%), showcasing the quality of `FedorAS`-produced bespoke architectures.

With respect to heterogeneous baselines (ZeroFL [66], FjORD[34]), we see that `FedorAS` consistently leads to models with higher accuracy across tiers that are in the respective clients' computational budget (Tab. 4). At the same time, we depict how `FedorAS` performs compared to FjORD and randomly selected architectures trained naively in a resource-aware manner in Fig. 3. One can clearly see that the degrees of architectural freedom that our solution offers leads to significantly better accuracy per resource tier. Notably, we perform 15.20% and 12.58% better on average than FjORD on CIFAR-10 and Shakespeare, respectively, while still respecting client eligibility.

While model accuracy may not seem to scale proportionally to their size, we attribute this to the limited participation eligibility of clients, an innate trait of system heterogeneous landscape. Normally, this can cause performance gaps due to limited exposure to federated data (Fig. 1). We argue that `FedorAS` is able to bridge this gap (+0.03 points (p) avg), by means of weight sharing and OPA, +1.72 p more effectively than FjORD (avg Tier 4 vs Tier 1 gap across datasets).

**Additional results.** Additional results on the convergence and fairness of `FedorAS` as well as alternative client allocation [44] to clusters are provided in Appendix G.2, I and H.3, respectively.

## 4.3 SUPERNET WEIGHT-SHARING WITH FEDORAS

Here, we evaluate the impact of weight sharing through `FedorAS`' supernet. We compare the performance of `FedorAS`' models to the same architectures models, but trained end-to-end in a federated manner, across all four datasets. With this comparison, we aim to showcase that `FedorAS` not only comes up with better architectures, but it also effectively transfers knowledge between tiers through its supernet structure and OPA aggregation scheme. In order to accomplish that, we first train with `FedorAS` across the three stages described in Sec. 3 (*supernet-init*). Subsequently, we take the output architectures from our system, randomly initialise them and train end-to-end in a federated manner, where each model trains across all eligible clients (*rand-init*). Results are presented in Tab. 5.

Indeed, models benefit from being initialised from a supernet across cases; this means that weight-sharing mitigates both the limited model capacity of the lower-tier and the limited data exposure of large models. The accuracy improvement is further magnified as non-IID-ness increases, leading up to +15 pp over *rand-init*. Results on different tasks of the same dataset presented in Appendix F.

---

[2] We annotate absolute performance difference as (percentage) points as (p)p and relative difference as %.

[3] 1.11 vs 0.16 GFLOPS, 11.4M vs 1.62M parameters, 4000 vs 850 global rounds (750 rounds of supernet training and 100 rounds of model finetuning)

**Table 5:** Models discovered by `FedorAS` benefit from weight sharing across tiers. Models resulted from the search stage in `FedorAS` and subsequently FL-finetuned are compared to models using the same architecture but trained end-to-end in an FL manner (i.e. randomly initialised, *rand-init*) on eligible clients.

| Dataset | Clients | Setting | Partitioning | Initialisation | Classes | Tier 1 | Tier 2 | Tier 3 | Tier 4 |
|---|---|---|---|---|---|---|---|---|---|
| CIFAR-10 (Acc. (%) ↑ is better) | 100 | Standard | $\text{IID}_{\alpha=1000}$ | Supernet | 10 | **89.40**±0.19 | **89.60**±0.15 | **89.64**±0.22 | **89.24**±0.29 |
| | | | | *rand-init* | | 89.05±0.17 | 87.84±0.38 | 86.18±0.38 | 81.27±0.81 |
| | | | $\text{non-IID}_{\alpha=1.0}$ | Supernet | 10 | 85.99±0.13 | **86.30**±0.41 | **86.34**±0.19 | **86.46**±0.32 |
| | | | | *rand-init* | | **87.12**±0.44 | 86.29±0.86 | 85.10±0.44 | 80.10±1.92 |
| | | | $\text{non-IID}_{\alpha=0.1}$ | Supernet | 10 | **81.01**±0.46 | **81.53**±0.29 | **80.64**±0.66 | **80.85**±0.28 |
| | | | | *rand-init* | | 70.61±2.16 | 70.30±1.90 | 68.29±0.49 | 64.87±1.48 |
| CIFAR-100 (Acc. (%) ↑ is better) | 500 | Standard | $\text{non-IID}_{\alpha=0.1}$ | Supernet | 100 | **45.25**±0.13 | **45.84**±0.18 | **45.42**±0.39 | **45.07**±0.71 |
| | | | | *rand-init* | | 36.30±0.96 | 39.26±1.21 | 39.06±1.32 | 36.77±1.32 |
| Speech Commands (Acc. (%) ↑ is better) | 2112 | Standard | *given* | Supernet | 12 | 80.19±1.78 | **80.47**±1.69 | **81.0**±1.58 | **80.56**±0.40 |
| | | | | *rand-init* | | **81.92**±1.32 | 79.94±0.84 | 78.57±1.42 | 79.36±1.67 |

| Dataset | Clients | Setting | Partitioning | Initialisation | Classes | Tier 1 | Tier 2 | Tier 3 | Tier 4 | Tier 5 |
|---|---|---|---|---|---|---|---|---|---|---|
| Shakespeare (Perplexity ↓ is better) | 715 | Standard | *given* | Supernet | 90 | **3.43**±0.01 | **3.39**±0.04 | **3.38**±0.03 | **3.40**±0.01 | **3.42**±0.01 |
| | | | | *rand-init* | | 3.44±0.03 | 3.50±0.02 | 3.47±0.08 | 3.52±0.07 | 3.60±0.04 |

## 4.4 ABLATION STUDY

Next, we measure the contribution of specific components of `FedorAS` to the quality and performance of these models. To this direction, we firstly compare the performance of our system's aggregation compared to naive averaging. Subsequently, we measure the impact of sending smaller subspaces to clients for supernet training. We provide indicative results on CIFAR-10$_{\alpha=0.1}$.

**OPA vs. naive averaging.** OPA compared to FedAvg leads to increased accuracy by +1.2, +0.9, +1.01 and +1.78 pp for tiers 1-4, respectively. More details in Appendix G.2.

**Subspace sampling size.** `FedorAS` samples the supernet before communicating it to a client. For CIFAR-10 ($\alpha = 1.0$) and when $B_{\Phi_{comm}}$ is set to 22.3M parameters (i.e. allowing sending whole supernet), `FedorAS` yields 85.48%, 86.73%, 86.50% average accuracies across tiers for full, $1/2$ and $1/4$ of the size of the search space, respectively. Thus, not only is the overhead of communication reduced with `FedorAS`, but convergence to the same level of accuracy is faster. We hypothesise that this is due a better balance in the exploration vs. exploitation trade-off. Details in Appendix G.2.

## 4.5 EVALUATION OF THE SEARCH PHASE

In order to assess the quality of models during search (Stage 2), so far we have needed a proxy dataset to evaluate different paths and rank them. We consider this as a set of examples that each application provider has centrally to measure the quality of their deployed models in production settings.

**Validation set size.** The size and representativeness of the centralised dataset might affect the quality of the search. To gauge the sensitivity of the end models to the underlying distribution of the validation set, we sample down the validation set, as a held-out portion of the clients datasets, to 20% and 50% of the original size. We find no noticeable impact on final model quality. More in App. J.1.

**Federated search.** There may be also cases where no such dataset can be centralised. To this direction, we test whether our search can operate under a federated setting with partial participation in order to faithfully rank the quality of models stemming from the supernet. In this setting, we have implemented a *federated version* of NSGA-II. Instead of candidate models being evaluated on the same validation set, they are stochastically evaluated on sampled federated client datasets [64]. We hypothesise it is possible to maintain faithful ranking of models compared to centralised evaluation if enough clients are leveraged to evaluate models, at the cost of increased communication cost. Furthermore, we expect the overall cost of achieving robust evaluation to increase as non-IID-ness and #clients increase, and the instantaneous cost of sending models to decrease over time, as NSGA-II converges to well-performing models (i.e. decreased diversity of models). Fig. 4 shows results for CIFAR10$_{\alpha=0.1}$, with extra results and details presented in Appendix J.2. Our experiments support the aforementioned conjectures. Noticeably, even under highly non-IID settings, we can attain faithful FE at a reasonable cost (4 rounds to Kendall-$\tau$>0.8 for the results presented in Fig. 4).

**Correlation with final accuracy.** We observe that the best models after Stage 2 are rather unlikely to be the best ones after Stage 3 (Kendall-$\tau$ 0.2-0.3). However, they tend to achieve decent results, suggesting they are a safe choice if we want to keep fine-tuning to the minimum. See App. J.3.

## 5 RELATED WORK

**Data and system heterogeneity.** Traditionally, works have focused on tackling the statistical data heterogeneity [72, 50, 35, 27, 54] or minimising the upstream communication cost [46, 42, 76, 29, 5],

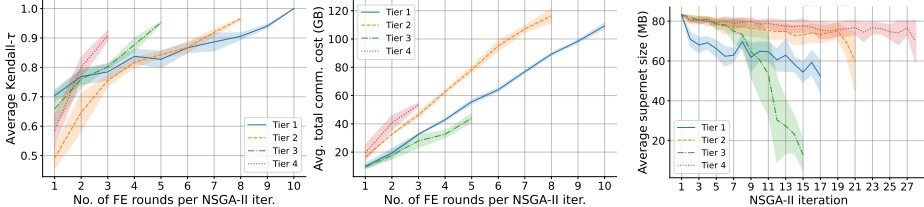

**Figure 4:** Ranking quality & cost of federated evaluation (FE) of models for CIFAR-$10_{\alpha=0.1}$ during federated search. Each time a new population of models is evaluated, a minimal supernet encompassing selected models is sent to a sample of clients: **left)** ranking correlation between scores produced by FE & centralised evaluation (CE), as a function of FE rounds ($\uparrow$ rounds = $\uparrow$ clients); **middle)** total communication cost of sending all necessary supernets to all clients, to run a full search; **right)** changes in the supernet size as NSGA-II progresses.

as the primary bottleneck in the federated training process. However, it has become apparent that computational disparity becomes an equally important barrier for contributing knowledge in FL. As such, lately, there has been a line of work focusing on this very problem which affects the convergence rate or fairness of the deployed system. Specifically, such solutions draw from efficient ML and attempt to dynamically alter the footprint of local models my means of structured (PruneFL [38], HeteroFL [24]), unstructured (Adaptive Federated Dropout [12], LotteryFL [47]) or importance-based pruning (FjORD [34]), quantisation (AQFL [2]), low-rank factorisation (FedHM [80]), sparsity-inducing training (ZeroFL [67]) or distillation (GKT [31]). However, each approach has limitations, either because they involve extra training overhead [31] and residence of multiple DNN copies in memory [2], or because they require specialised hardware for performance gains ([67, 12]). Last, some of the architectural changes proposed may not offer the degrees of freedom that NAS exposes [34, 24, 80, 47] and perform uni-dimensional search over the control variable across layers [34, 24], or may have a different optimisation objective altogether [47].

**Federated NAS.** The concept of performing NAS in a federated setting has been considered before [30, 60, 81, 84, 55]. However, current solutions vary greatly in their approaches and goals, with most of them being applied and applicable to the cross-silo setting, where full participation and small number of participants are expected. Specifically, one of the first works in the area was FedNAS [30], which adopts a DARTS-based approach and aims to find a globally good model for all clients, to be personalised at a later stage. This approach requires the whole supernet to be transmitted and kept in memory for architectural updates, which leads to excessive requirements (Fig. 2) that make it largely inapplicable for cross-device setups with clients of limited capabilities. To mitigate this requirement, [81] proposes an RL-based approach for cross-silo FL-based NAS. Despite the intention, it still incurs significant overheads due to RL-based model sampling convergence and single model training per client. A somewhat different approach is adopted by HAFL [55], which leverages graph hypernetworks as a means to generate outputs of a model. While interesting, performance and scalability are not on par with current state-of-the-art. On the front of personalisation, FedP-NAS [32] searches for architectures with a shared based component across clients and a personalised component. However, this work is only aimed at IID vision tasks and involves a meta-step for personalisation, which increases training overheads significantly. At the other extreme for cross-silo personalised FL, SPIDER [60] aims at finding personalised architectures per client. It requires the whole space on-device and federation is accomplished through a second static on-device network. These non-negligible overheads make porting this approach to the cross-device setting non-trivial.

In contrast, `FedorAS` brings Federated NAS to the cross-device setting and is designed with resource efficiency and system heterogeneity in mind. This way, communication and computation cost is kept within the defined limits, respecting the runtime on constrained devices. Crucially, our system does not assume full participation of clients and is able to efficiently exchange knowledge through the supernet weight sharing and generalise across different modalities and granularity of tasks.

## 6 CONCLUSION

In this work, we have presented `FedorAS`, a system that performs resource-efficient federated NAS in the cross-device setting. Our method offers significantly lower overhead compared to existing federated NAS techniques, while achieving state-of-the-art performance compared to heterogeneous FL solutions, by means of effective weight sharing and flexible resource-aware training.

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

# Table of Contents

## A  INTRODUCTION

This Appendix extends the content of the main paper by providing support material to the study presented in Sec. 3, 4 and, additional insights about `FedorAS`. Concretely, the Appendix is divided into three main blocks of extra content:

- **Impact and Limitations.** We concisely present the broad impact and limitations of our work as well as future research directions in Sec. B, C.

- **Experimental Setup.** Sec. D provides details on the libraries used to build `FedorAS`, the datasets considered for experiments as well as the hyperparameters used to obtain the results presented in Section 4. Crucially, we provide a detailed description of the search spaces utilised in for each dataset domain, how tiers are defined and, a concise description of each baseline method included in this work.

- **Additional Experiments.** Sec. E.1 through I study different aspects of `FedorAS` such as: *i)* learning multiple tasks using the supernet in Sec. F; or *ii)* the convergence behaviour in Sec. G.2; *iii)* the fairness aspect of `FedorAS`; or under new scenarios altogether, such as: *iv)* an alternative procedure to assign clients to tiers in Section H.3; or *iv)* alternative search methods for Stage 2 in Sec. J.

Overall, the following content substantially extends what is already presented in the main text.

## B    BROADER IMPACT

Our system, `FedorAS`, performs federated and resource-aware NAS in the cross-device setting. Despite the benefits illustrated compared to centralised NAS and cross-silo FL solutions, running Neural Architecture Search is still a resource-demanding process, in terms of compute, memory and network bandwidth. While `FedorAS`'s target devices can be of significantly lower TDP (i.e. smartphones and IoT devices vs. server-grade GPUs) – with consequences on the overall energy consumption of training [67] – they are resources not directly owned by the operator. As such, special care needs to be taken with respect to how many device resources are leveraged at any point in, so as not hinder the usability of the device or invoke monetary costs to the user [41].

## C    LIMITATIONS & FUTURE WORK

Despite the challenges addressed by `FedorAS`, our prototype has certain limitations. First and foremost, we have opted to cluster devices (i.e. in tiers) based on their FLOPS. While it is perfectly normal to divide clusters based on other criteria (e.g. memory, latency, energy) or in a multi-objective manner, we have kept it simple. Moreover, one can opt for biased sampling of i) clients participating in a round, ii) the subspace they get communicated and iii) the paths sampled from that subspace, we opted for uniform sampling for all of the above for simplicity, uniform coverage of the search space and fairness in participation. We leave the exploration of such strategies as future work. Last but not least, we have considered privacy-enhancing techniques, such as Differential Privacy [14] or Secure Aggregation [11, 6], as orthogonal to our scheme. Combining Federated NAS with such strategies can expose interesting trade-offs of exploration-exploitation-privacy budgets that could be explored in the future.

## D    DETAILED EXPERIMENTAL SETUP

### D.1    IMPLEMENTATION

`FedorAS` was implemented on top of the Flower (v0.18) [9] framework and PyTorch (v1.11.0) [63]. We run all our experiments on a private cloud cluster in a simulated manner, across four iterations each and report averages and standard deviations.

### D.2    DATASETS

We partition CIFAR-10/100 following the Latent Dirichlet Allocation (LDA) partitioning [36], with each client receiving approximately equisized training sets. For CIFAR-10, we consider $\alpha \in \{1000, 1.0, 0.1\}$ configurations, while for CIFAR-100, we adopt $\alpha = 0.1$ as in [70]. The remaining datasets come naturally partitioned.

**CIFAR-10/100.** The CIFAR-10 datasets contains 10k and 50k 32x32 RGB images in its test and training sets respectively comprising ten classes. The goal is to classify these images correctly. Similarly, CIFAR-100 follows an identical partitioning but, this time, across 100 classes (*fine* lebels) or 20 superclasses (*coarse* labels). Both CIFAR datasets have a uniform coverage across their classes.

**SpeechCommands.** The Speech Commands datasets [77] is comprised of 105,829, 16KHz 1-second long audio clips of a spoken word (e.g. "yes", "up", "stop") and the task is to classify these in a 12 or 35 classes setting. The datasets comes pre-partitioned into 35 classes and in order to obtain the 12-classes version, the standard approach [8, 19, 75] is to keep 10 classes of interest (i.e. "yes", "no", "up", "down", "left", "right", "on", "off", "stop", "go"), place the remaining 25 under the "unknown"

class and, introduce a new class "silence" where no spoken word appear is the audio clip. In this work we consider SpeechCommandsV2, the most recent version of this dataset. The dataset spans three disjoint set of speakers: 2112 form the training set, 256 for validation and, 250 for testing. In `FedorAS`, the supernet training phase makes uses of the 2112 clients in the training partition only. The results are obtained by measuring the performance of the discovered models on the data of the 250 clients comprising the test set. Data is not uniformly distributed and some clients have more data than others. This dataset is processed by first extracting MFCC [21] features from each audio clip [86, 8]. Across our experiments we extract 40 MFCC features from a MelSpectrogram where each audio signal is first sub-sampled down to 8KHz and then sampled using 40ms wide time windows strided 20ms appart. This results in 1-second audio clip being transformed into a $1\times40\times51$ input that can be passed to a CNN.

**Shakespeare.** This dataset is built [17, 58] from *The Complete Works of William Shakespeare* and partitioned in such a way data that from each role in the play is assigned to a client. This results in a total of 1,129 partitions where the average number of samples per device is 3.7K and the standard deviation 6.2K samples. This makes Shakespeare a relatively imbalanced dataset. The task is to correctly predict the next character in a dialog given the previously seen characters in the sentence. The vocabulary considered has 86 English characters as well as four special tokens: start and end of sentence, padding and out-of-vocabulary tokens.

### D.3  SEARCH SPACES

We assume that the largest model in the search space is the maximal size that any of the clients can handle. The minimum cost is set to the fixed cost of the network (i.e. cost of non-searchable components). Given this range, and number of clusters $C$, we define the first and last clusters at $prct_l, prct_r$ percentiles of a randomly sampled set of models from the search space and linearly scale the $C - 2$ clusters in-between. All supernet search spaces are depicted in Fig. 5 and described below.

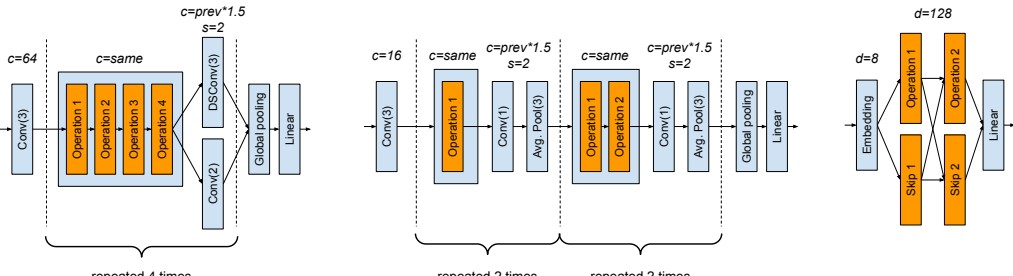

**Figure 5:** Summary of models used in our experiments: **left**) CIFAR-10 and CIFAR-100, **middle**) Speech Commands, **right**) Shakespeare. Blocks highlighted in blue are fixed, orange blocks represent searchable layers. c – output channels, s – stride, d – output feature dimension, DSConv – depthwise separable convolution. Whenever a layer has more than one input they are added. All convolutions are followed by BN and ReLU. For convolution and pooling operations, numbers in parentheses represent window sizes.

**CIFAR-10 and CIFAR-100.** We use a ResNet-like search space similar to the one used in, for example, Cai et al. [16]. Specifically, our model is a feedforward model with a fixed (*i.e.* non-searchable) stem layer followed by a sequence of 3 searchable blocks, each followed by a fixed reduction block. A standard block consists of 4 searchable layers organized again in a simple feedforward manner. Operations within a standard block preserve shape of their inputs, but are allowed to change dimensions of intermediate results. On the other hand, the goal of reduction blocks is to reduce spatial resolution ($2\times$ in each dimension) and increase number of channels ($1.5\times$). Reduction blocks are fixed throughout and consists of a depthwise separable convolution $3 \times 3$, with the first part performing spatial reduction and the second increasing the number of channels, and a standard $2 \times 2$ convolution applied to a residual link. The sequence of blocks is finished with a global average pooling per-channel and a classification layer outputting either 10 or 20/100 logits, which is the only difference between the two models. Each convolution operation is followed by a batch normalization (BN) and ReLU.

In our experiments we considered the following candidate operations:

- standard convolution $1 \times 1$ with BN and ReLU

- depthwise separable convolution $3 \times 3$ with expansion ratio (controlling the number of intermediate channels) set to $\{0.5, 1, 2\}$, with each expansion ratio being an independent choice in our search space
- MobileNet-like block consisting of a convolution with kernel size $k$ and expansion ratio $e$, followed by a Squeeze-and-Excitation layer, followed by a $1 \times 1$ convolution reverting the channels expansion, we considered $\{(k = 1, e = 2), (k = 3, e = 0.5), (k = 3, e = 1), (k = 3, e = 2)\}$
- identity operation

The stem layer was set to a $3 \times 3$ convolution outputting 64 channels.

**Speech Commands.** The model follows the one used for the CIFAR datasets but is made more lightweight – to roughly match what can be found in the literature – by reducing stem channels to 16 and including only 1 (resp. 2) searchable operations in the first (resp. last) two blocks. Additionally, reduction block only includes a single $1 \times 1$ convolution, that changes the number of channels, followed by a $3 \times 3$ average pooling that reduces spatial dimensions. We also include additional candidate operations:

- standard convolution $3 \times 3$
- depthwise separable convolution with kernel size 1 and expansion ratio 2

All other operations from the CIFAR model are also included.

**Shakespeare.** We base our design on the model used by FjORD. [34]. Specifically, the model is a recurrent network that begins with a fixed embedding layer outputting 8-dimensional feature vector per each input character. This is then followed by a searchable operation #1 that increases dimensionality from 8 to 128; in parallel, we have a searchable skip connection #1 operation whose output is added to the output of the operation #1. Later there is another searchable operation #2 with its own skip connection #2, both keeping the hidden dimension at 128. Again, their outputs are added and passed to the final classification layer.

Candidate operations for each of the four searchable layers are mainly the same, with minor adjustments made to make sure a valid model is always constructed. These include:

- an LSTM layer [33],
- a GRU layer [18],
- a LiGRU layer with `tanh` activation [68],
- a QuasiRNN layer [13],
- a simple Linear layer (no recurrent connection) followed by a `sigmoid` activation,
- a 1D convolution with kernel 5 spanning time dimension (looking at the current character and 4 previous ones), followed by a `sigmoid` activation (no normalisation),
- identity operation (only included in later operation, after feature dimension has been increased to 128),
- zero operation, outputting zeros of the correct shape (only included in skip connection layers).

For LiGRU and QuasiRNN we used implementations provided by the Speechbrain project [69], after minor modifications. For others, we used standard operations from PyTorch. All operations were used as unidirectional. We did not use any normalisation throughout the model.

### D.4 TIERS: DEFINITION AND CLIENT ASSIGNMENT

With `FedorAS` the discovery and training of architectures happens in a tier-aware fashion as a federated process. In this work we considered splitting each search space along the FLOPs dimensions (but other splits are possible, e.g: energy, peak-memory usage, etc – or a combination of these). Fig. 6 illustrates the span in terms of model parameters and FLOPs of each search space presented in D.3 and the split along the FLOPs dimensions for each of them. These search spaces vary considerably in

**Table 6:** This table summarises how each search space is split into tiers. Parameters $\rho_L$ and $\rho_H$ are used to conveniently split the FLOPs axis for each dataset and present challenging scenarios for FedorAS. The last column refers to the percentage of the total clients that are assigned to each tier.

| Dataset | $[\rho_L, \rho_H]$ | Tier | FLOPs range | # Models | Portion(%) | Clients (%) |
|---|---|---|---|---|---|---|
| CIFAR-10 | [0.0, 0.95] | T1 | [0, 120.9M] | $239.4 \cdot 10^{12}$ | 12.92% | 25% |
| | | T2 | (120.9M, 223.2M] | $1089.4 \cdot 10^{12}$ | 58.80% | 25% |
| | | T3 | (223.2M, 325.4M] | $477.2 \cdot 10^{12}$ | 25.75% | 25% |
| | | T4 | (325.4M, 716.0M] | $46.9 \cdot 10^{12}$ | 2.53% | 25% |
| CIFAR-100 | [0.0, 0.9] | T1 | [0, 111.5M] | $168.4 \cdot 10^{12}$ | 9.09% | 25% |
| | | T2 | (111.5M, 204.3M] | $975.4 \cdot 10^{12}$ | 52.64% | 25% |
| | | T3 | (204.3M, 297.1M] | $609.5 \cdot 10^{12}$ | 32.89% | 25% |
| | | T4 | (297.1M, 716.0M] | $99.7 \cdot 10^{12}$ | 5.38% | 25% |
| CIFAR-100 (multi-task setting of Appendix F) | [0.0, 0.9] | T1 | [0, 111.5M] | $168.4 \cdot 10^{12}$ | 9.09% | 50% |
| | | T2 | (111.5M, 204.3M] | $975.4 \cdot 10^{12}$ | 52.64% | 25% |
| | | T3 | (204.3M, 297.1M] | $609.5 \cdot 10^{12}$ | 32.89% | 12.5% |
| | | T4 | (297.1M, 716.0M] | $99.7 \cdot 10^{12}$ | 5.38% | 12.5% |
| SpeechCommands | [0.3, 0.925] | T1 | [0, 5.0M] | $827.5 \cdot 10^{3}$ | 46.71% | 80% |
| | | T2 | (5.0M, 7.5M] | $617.4 \cdot 10^{3}$ | 34.85% | 1.25% |
| | | T3 | (7.5M, 10.1M] | $260.2 \cdot 10^{3}$ | 14.69% | 1.25% |
| | | T4 | (10.1M, 20.0M] | $66.4 \cdot 10^{3}$ | 3.75% | 17.5% |
| Shakespeare | [0.1, 0.77] | T1 | [0, 7.8M] | 316 | 13.48% | 20% |
| | | T2 | (7.8M, 12.8M] | 577 | 24.54% | 20% |
| | | T3 | (12.8M, 17.8M] | 787 | 33.48% | 20% |
| | | T4 | (17.8M, 22.8M] | 473 | 20.14% | 20% |
| | | T5 | (22.8M, 33.8M] | 196 | 8.37% | 20% |

terms of size and span, which motivated us to use different number of tiers or client-to-tier assignment strategies. Tab 6 shows the the FLOPs ranges considered for each tier, the number of models in those sub-spaces exposed to FedorAS as well as the ratio of models in the entire search space that fall onto each tier.

**Identifying FLOPs ranges for each tier.** Regardless of the dataset, we follow a common approach to divide the FLOPs dimension that each dataset spans. The main aim is to finely control how many models fall onto the smallest/largest tier, given that these are considerably sparse regions of the entire search space. If we were to evenly split the FLOPs dimension, almost no architecture would fall onto the largest tier. To simplify the process of defining where each tier's boundaries lie, we follow these steps which require two hyperparameters: (1) we construct a long array of FLOPs from models sampled from the search space and this array is then sorted from lowest to highest; then, (2) we normalise this array of FLOPs and compute the cumulative sum of such normalised array; finally, (3) we identify the lower limit of the highest tier as maximum FLOPs value found in the first $\rho N, \rho \in [0, 1]$, elements of the array where $N$ is the number of samples taken (we found 100K to work well). Essentially, we find the FLOPs for which the approximated PDF over the FLOPS of a given search space doesn't surpass $\rho$. Once that FLOPs value is identified, the FLOPs range is evenly split for the tiers below and the higher limit for the largest tier becomes the maximum FLOPs a model in the search space can have. This is the approach for CIFAR-10 and CIFAR-100 as illustrated in Fig. 6 (a) and (b). For SpeechCommands and Shakespeare we introduce a similar approach to give a wider range to the smallest tier. We refer to $\rho_L$ and $\rho_H$ as the PDF ratios to identify the boundaries splitting Tier1&2 and Tier3&4 respectively. The concrete values for each dataset as well as the FLOPs values for each tier boundary are shown in Tab 6.

**Clients to tiers assignment.** For CIFAR-10/100 clients are uniformly assigned to a cluster of devices or tier. For these datasets we consider four tiers, so each ends up containing 25% the clients resulting in 25 clients for CIFAR-10 and 125 clients for CIFAR-100. Similarly, for Shakespeare clients are also uniformly assigned to a tier, resulting in 143 clients per tier. For SpeechCommands, we designed a more challenging setup and divide the clients into tiers as follows: 80% of clients are assigned to be tier-1 devices, 17.5% are Tier-4 devices and the rest is evenly split into Tier-2 and Tier-3. This distribution better represents the the types of systems in the wild that perform keyword-spotting [86], where the majority of the commercially deployed systems run these applications on low end CPUs or microcontrollers due to their *always-on* nature. Across datasets, the client-to-tier assignments was done irrespective of the amount of data these contained or distribution over the labels. An alternative client to tier allocation is provided in Sec. H.3.

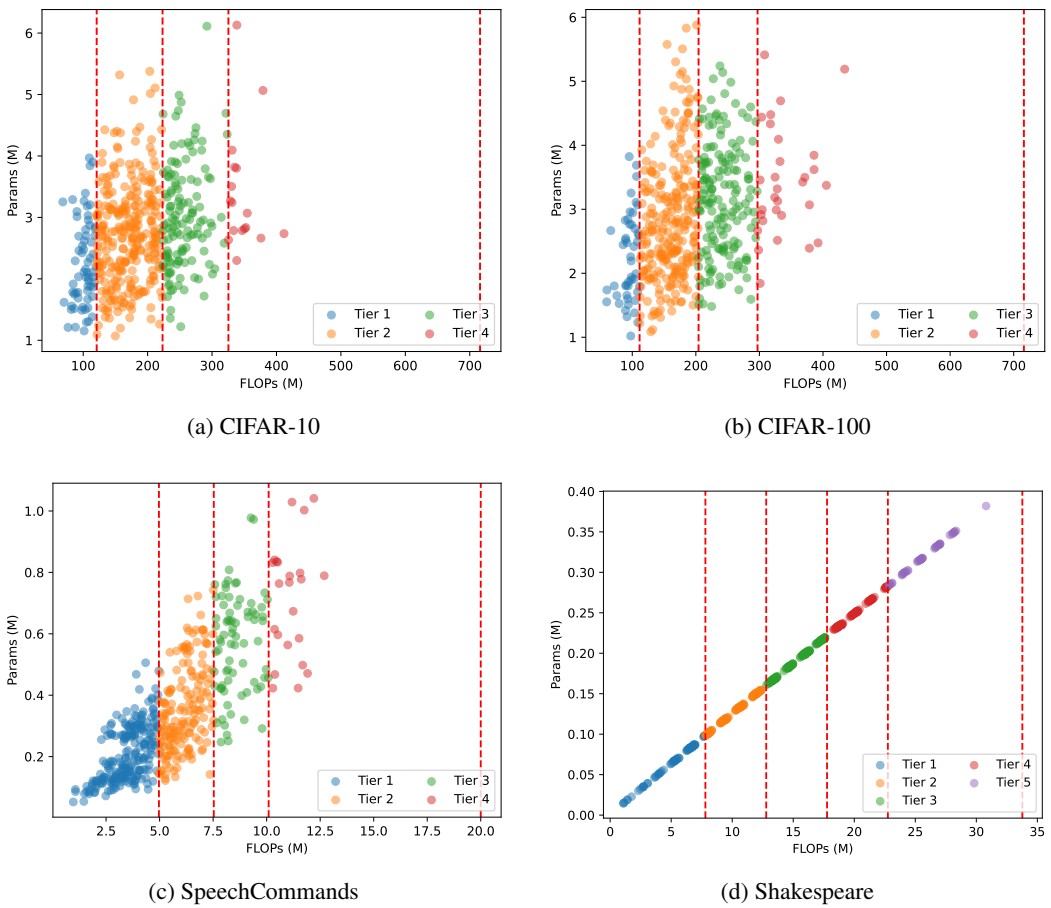

(a) CIFAR-10

(b) CIFAR-100

(c) SpeechCommands

(d) Shakespeare

**Figure 6:** For each search space, we randomly sample 500 architectures and color code them based on the tier they belong to. Vertical dashed lines represent the boundaries between device tiers. For Shakespeare, the majority of the candidate operators in the searchspace include just linear layers or no-op layers, as a results the number of FLOPs grows almost linearly with the model of parameters.

### D.5 HYPERPARAMETERS

Here we present the hyperparameters used across all datasets and tasks to generate the results presented in Sec. 4. In Tab. 7 we show the hyperparamters utilised for the first stage in the `FedorAS` framework: federated training of the supernet. Noticeably we require more local epochs in this stage as training each client effectively (pre-)trains multiple models and we need sufficiently many forward passes in order to sample a large enough number of paths on each client. In the context of `FedorAS`, the combination of batch size and number of local epochs define the exploration vs. exploitation trade-off in the communicated subspace. It is natural that if not enough paths are explored or if paths overfit the local data, the overall NAS results and thus model ranking would be severely affected. As such, we increase the number of local epochs together with a batch size, to ensure that the total number of forward passes remains constant for the same number of training examples on a client. After this stage, the best model for each tier is extracted from the supernet and then, they get finetuned in a per-tier aware manner (i.e. clients in tier $T$ and above can finetune a model that belong to tier $T$). The hyperparameters of these two consecutive stages are shown in Tab. 8.

Regarding the *rand-init* results (i.e. models discovered by `FedorAS` but trained from scratch – discarding the weights given by the supernet) we present the hyperparameters in Tab. 9. These hyperparameters were the ones used to generate the results in Tab. 5 and 13 as well as Fig. 3. In Sec. 4.2, we present a CIFAR-100 result that largely outperforms existing federated baseline of [70]. This was achieved by a model discovered by `FedorAS` but trained in the *rand-init* setting following

**Table 7:** Hyperparameters used for the federated supernet training stage in `FedorAS`, as described in Sec. 3.1. The learning rate is kept fixed during this stage. In all datasets, the aggregation strategy followed the proposed histogram-informed methodology OPA, first presented in Sec. 3.1.

| Dataset | # Federated Rounds | # Clients per round | # Local Epochs | Local Optimizer | Batch Size | LR | Momentum | Gradient Clipping |
|---|---|---|---|---|---|---|---|---|
| CIFAR-10 | 500 | 10 | 50 | SGD | 128 | 0.1 | 0.9 | N |
| CIFAR-100 | 500/750 | 10 | 25 | SGD | 64 | 0.1 | 0.9 | N |
| SpeechCommands | 750 | 21 | 25 | SGD | 64 | 0.1 | 0.9 | N |
| Shakespeare | 500 | 16 | 5 | SGD | 4 | 1.0 | 0.0 | Y |

**Table 8:** Hyperparameters used by `FedorAS` to search for the best model in the supernet and finetune them in a tier-aware fashion as described in Sec. 3.2 and Sec. 3.3 respectively. Searching iterations shown are allocated per-tier (i.e. CIFAR-10 has 4 tiers so a total of 4K valid models would be considered during the search). *Cosine* LR scheduling gradually reduces the initial LR (shown in the table) by an order of magnitude over the span of the the the finetuning process. For Shakespeare, *step* LR decay worked best. This is applied at rounds 50 and 75, each decreasing the LR by a factor of $10\times$. SpeechCommands assigns different search iterations based on the ratio of clients assigned to each tier. `FedorAS` scales the amount of search for valid models within the tier accordingly. For Shakespeare, the sub-searchspaces that yield Tier-1 and Tier-5 models are smaller than for the other tiers, we therefore consider fewer search iterations.

| Dataset | # Tiers | # Search Iterations | # Finetune Rounds | # Clients per round | Local Epochs | Batch Size | LR | LR Scheduling | Other Hyperparams |
|---|---|---|---|---|---|---|---|---|---|
| CIFAR-10 | 4 | 1000 | 100 | 6 | 1 | 32 | 0.01 | cosine | momentum=0.9 |
| CIFAR-100 | 4 | 1000 | 100 | 6 | 1 | 32 | 0.01 | cosine | momentum=0.9 |
| SpeechCommands | 4 | 500/2000 | 100 | 21 | 1 | 32 | 0.01 | cosine | momentum=0.9 |
| Shakespeare | 5 | 100/150 | 100 | 16 | 1 | 4 | 1.0 | step | g.clipping=5 |

the setup as in [70]: 10 clients per round for 4k rounds using batch 20, starting learning rate of 0.1 decaying to 0.01 following a *cosine* scheduling.

### D.6 CROSS-DEVICE FEDERATED NAS EVALUATION

In Tab. 2 of Sec. 4.2 we compared `FedorAS` against FedNAS in the cross-sile setting with 100 clients and 10 clients randomly sampled on each round. Both methods follow a substantially different approach as far as NAS is concern and, as a result, FedNAS has a significantly higher memory peak than FedorAS for the same batch size. For example, for a batch size of 64 images, FedorAS sees a memory peak of 998MB whereas FedNAS requires 7674MB. Similarly, both methods translate in different compute footprints for each client. For example, FedNAS requires on average 716 GFLOPs per client (assuming each client), while clients in FedorAS need 280 GFLOPs for the same amount of local epochs and data in the client. Due to these differences, we aimed at normalising these aspects to make the comparison fair. These results are shown in Tab. 10, which extends the content of Tab. 2.

### D.7 BASELINES

In this section we faithfully describe what each baseline represents in the experiments of the main paper and the appendix to clarify with what we are comparing in each section.

**Tier-unaware** (Fig.3): This baseline represents model architectures that have been trained end-to-end in a federated manner without any awareness of client eligibility. We are using FedAvg with hyperparameters similar to those presented in Tab. 9.

**Tier-aware** (Fig.3): This baseline adds client eligibility awareness to the previous baseline. This means that models of certain footprint can be trained only on clients of the eligible cluster and above.

**FjORD** [34] (Fig.3, Tab. 4): FjORD is a baseline that is tackling system heterogeneity by means of Ordered Dropout. It assumes a uniform dropout rate across layers, essentially keeping the control variable one-dimensional and thus offering fewer degrees of architectural freedom. Nevertheless, it enables the dynamic extraction of candidate submodels without the need to retrain or finetune.

**Table 9:** The architectures discovered by `FedorAS` can also be trained from scratch. This table contains the hyperparameters utilised to generate the *rand-init* results shown in Tab. 5 and 13. Training of these baselines is also performed in a tier-aware fashion. *Cosine* LR scheduling gradually reduces the initial LR (shown in the table) by an order of magnitude over the span of the the finetuning process. For Shakespeare, *step* LR decay worked best. This is applied at rounds 250 and 375, each decreasing the LR by a factor of $10\times$.

| Dataset | # Federated Rounds | # Clients per round | Local Epochs | Batch Size | LR | LR Scheduling | Other Hyperparams |
|---|---|---|---|---|---|---|---|
| CIFAR-10 | 500 | 10 | 1 | 32 | 0.1 | cosine | momentum=0.9 |
| CIFAR-100 | 500 | 10 | 1 | 32 | 0.1 | cosine | momentum=0.9 |
| SpeechCommands | 500 | 21 | 1 | 32 | 0.1 | cosine | momentum=0.9 |
| Shakespeare | 500 | 16 | 1 | 4 | 1.0 | step | g.clipping=5 |

**Table 10:** Cross-device federated NAS on CIFAR-10. We compare `FedorAS` and FedNAS while normalising aspects that are critical to on-device training: namely the number of FLOPs clients do in a given round and the memory peak seen over the course of such training. The later directly impacts on which devices a particular model could be trained on. We normalise memory peak by lowering the batch size that FedNAS uses (from 32 to 16) and reduce the number of local epochs in `FedorAS` down to 20 to match the FLOPs of typical FedNAS client. For further context, we maintain the results of `FedorAS` with 50 local epochs – the setting used throughout the majority of the experiments in this paper.

| Dataset | Method | Local Epochs | Batch | Mem. Peak (GB) | GFLOPs/client | Perf. (%) |
|---|---|---|---|---|---|---|
| CIFAR10$_{\alpha=1}$ | FedNAS | 10 | 32 | 3837 | 1431 | **90.02** |
| | FedNAS | 10 | 16 | **1919** | 1431 | 85.45 |
| | FedorAS | 20 | 128 | 1996 | **1402** | 87.21$\pm$0.15 |
| | FedorAS | 50 | 128 | 1996 | 3504 | 86.46$\pm$0.32 |
| CIFAR10$_{\alpha=0.1}$ | FedNAS | 10 | 32 | 3837 | 1431 | 65.28 |
| | FedNAS | 10 | 16 | **1919** | 1431 | 54.84 |
| | FedorAS | 20 | 128 | 1996 | **1402** | 79.41$\pm$0.31 |
| | FedorAS | 50 | 128 | 1996 | 3504 | **81.53**$\pm$0.29 |

This particular variant is assuming the experimental setup of the original paper which shards the CIFAR-10 dataset per client without LDA. The Shakespeare setup remains the same as ours.

**FjORD-LDA** [34] (Fig.3, Tab. 4): For this baseline, we implemented FjORD and ran it on the same number of clusters as `FedorAS`, with LDA for CIFAR-10.

**FedNAS** [30] (Tab. 2, 3): FedNAS was one of the first papers attempting to perform NAS in a federated setting. Its search is DARTS-based and it was primarily aimed for the cross-silo setting. Cross-device FL NAS of limited scale (20 total clients with 4 clients sample per round) was also showcased in the original paper. We adjusted the code of FedNAS to perform cross-silo and cross-device FL NAS with setups of varying training footprints by adjusting the amount of local epochs and batch size. We have kept the original search space.

**SPIDER** [60] (Tab. 3): SPIDER is another paper performing personalised Federated NAS in the cross-silo setting. As there is no publicly available codebasse, we assume their setting with `FedorAS` and compare on CIFAR-10.

**ZeroFL** [66] (Tab. 4): For this baseline, we borrow the results of respective paper for CIFAR-10$_{\alpha=\{1,1000\}}$, which assumes the same setup as ours. We present the results for sparsity level of 90% and annotate its footprint as the original model FLOPS and the number of non-zero parameters.

**Oort** [44] (Tab. 4): The Oort framework proposes a participation sampling criterion by which clients are sampled based on their utility (i.e. how much their data can contribute to the global model) while also taking the device capabilities into consideration. Over the course of FL training, the sampling of clients with high data and system utility is prioritised to reduce the time needed to convergence. For SpeechCommands (see Tab. 4), Oort made use of a ResNet-34 model.

**PyramidFL** [49](Tab. 4): At a high-level, PyramidFL proposes a framework similar to that in Oort. The core difference between these two methods is that PyramidFL leverages more fine-grained

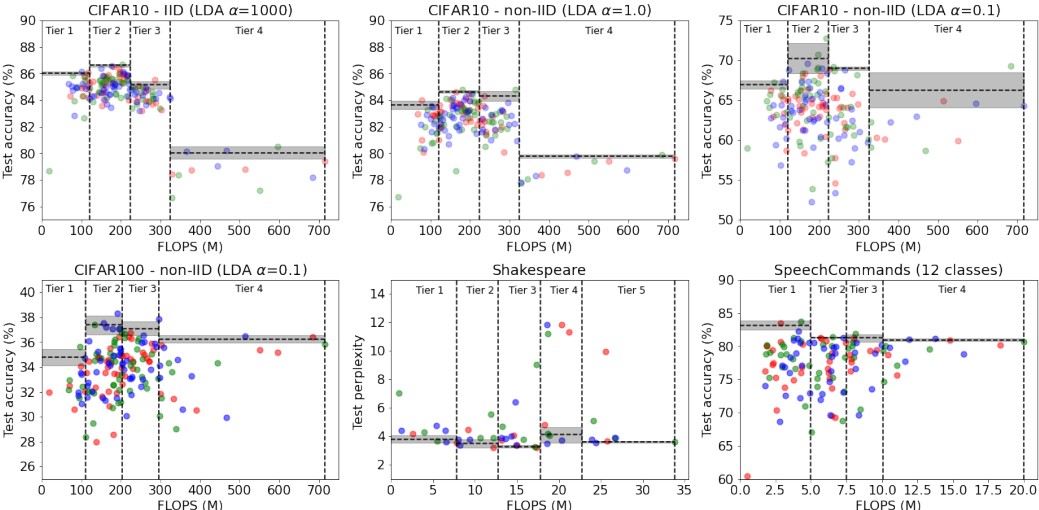

**Figure 7:** For each dataset, we randomly sample their search space and train these models end-to-end. We repeat this process (with new samples) three times, and overlap the scatter plots of different runs (i.e. red, green, blue). Then, for each tier we average the best performing models across each of the three runs and plot the result as a dashed horizontal line with grey area representing $\pm$ standard deviation. The compute budget allocated to generate the data for each plot is equivalent to the cost of running `FedorAS`, which also yields one model per tier.

statistics when assessing the contribution potential (i.e. utility) of the selected clients. It also uses ResNet-34 for SpeechCommands.

**rand-init** (Tab. 5, 13, 17): This baseline refers to the concept of running the search, coming up with an architecture and subsequently re-initialising the weights and running a conventional federated training setup. It shows vanilla scaling that would be obtained if we simply trained identified models in a standard way, using random initialization and full FL training using standard practices. Model architectures are kept the same between `FedorAS` and "*rand-init*" experiments. Models belonging to higher tiers, as a whole, are never trained on data belonging to devices from lower tiers.

# E    COMPARISON WITH ALTERNATIVE NAS ALGORITHMS

Note: this section considers replacing `FedorAS` as a whole with a different approach to searching for the best performing model in the same search space etc. On the other hand, for additional details about the searching stage of `FedorAS` (Stage 2) please see Section J.

## E.1    COMPARISON AGAINST RANDOM SEARCH

In the main paper we compare our `FedorAS` to the state-of-the-art methods from the literature (Sec. 4.2) and also investigate the impact of initializing models with weights from a supernet (Sec. 4.3). Here we present one more important baseline for any NAS algorithm - comparison to a random search. Specifically, while we have already shown that models found by `FedorAS` in most cases benefit from supernet training, this does constitute a conclusive argument justifying the necessity of the entire process in a broader context. Perhaps there exist models that do not need to be initialized from a supernet but, since the focus of our work is on supernet training, we missed them due to biased conditioning?

In order to check this hypothesis we run a simple random search algorithm, to get an estimate of the expected *best case* performance of models from our search spaces when trained following standard FL procedure. This simple baseline was allowed to train random models until the total training cost exceeded the cost of running `FedorAS` – this turned out to be equivalent to roughly 40 fully-trained models. After we run out of the training budget, we simply get the best model for each tier as our solution to NAS. We repeated the entire process 3 times for each search space and report average and

**Table 11:** Comparison of performance of models found with `FedorAS` and with a simple random search.

| Dataset | FedorAS | | | | | Random search | | | | |
|---|---|---|---|---|---|---|---|---|---|---|
| | T1 | T2 | T3 | T4 | T5 | T1 | T2 | T3 | T4 | T5 |
| CIFAR-10 ($\alpha = 1000$) | **89.40**±0.19 | **89.60**±0.15 | **89.64**±0.22 | **89.24**±0.29 | - | 86.03±0.13 | 86.39±0.28 | 85.18±0.26 | 79.84±0.75 | - |
| CIFAR-10 ($\alpha = 1.0$) | **85.99**±0.13 | **86.30**±0.41 | **86.34**±0.19 | **85.58**±0.55 | - | 83.68±0.25 | 84.50±0.31 | 84.30±0.38 | 79.65±0.18 | - |
| CIFAR-10 ($\alpha = 0.1$) | **81.01**±0.46 | **81.53**±0.29 | **80.64**±0.66 | **80.85**±0.28 | - | 66.75±0.74 | 70.85±1.53 | 69.00±0.18 | 66.08±2.29 | - |
| CIFAR-100 | **45.25**±0.13 | **45.84**±0.18 | **45.42**±0.39 | **45.07**±0.71 | - | 34.41±0.90 | 37.64±0.46 | 36.98±0.64 | 35.45±0.90 | - |
| SpeechCommands | 80.19±1.78 | 80.47±1.69 | 81.00±1.58 | 80.56±0.40 | - | 82.84±1.08 | 81.17±0.26 | 81.10±0.74 | 80.61±0.61 | - |
| Shakespeare | **3.43**±0.01 | **3.39**±0.04 | **3.38**±0.03 | **3.40**±0.01 | **3.42**±0.01 | 3.86±0.33 | 3.54±0.28 | 3.45±0.32 | 3.80±0.30 | 3.60±0.39 |

**Table 12:** Comparison of `FedorAS` vs centralised NAS setting.

| Method | Dataset | Tier 1 | Tier 2 | Tier 3 | Tier 4 |
|---|---|---|---|---|---|
| Centralised NAS | CIFAR-10$_{centralised}$ | 93.77±0.20 | 94.14±0.34 | 94.04±0.22 | 94.33±0.09 |
| FedorAS | CIFAR-10$_{\alpha=1000}$ | 89.40±0.19 (**-4.37**) | 89.60±0.15 (**-4.54**) | 89.64±0.22 (**-4.40**) | 89.24±0.29 (**-5.09**) |

**Table 13:** Models discovered by `FedorAS` benefit from weight sharing across tiers compared to models using the same architecture but trained end-to-end on clients that support them. Models derived from a `FedorAS` supernet outperform their baselines by large margins in most cases. First part: accuracy, second part: perplexity.

| Dataset | Clients | Setting | Partitioning | Mode | Classes | Tier 1 | Tier 2 | Tier 3 | Tier 4 |
|---|---|---|---|---|---|---|---|---|---|
| CIFAR-100 | 500 | Multi-task | non-IID$_{\alpha=0.1}$ | FedorAS *rand-init* | $[20, 20, 100, 100]$ | **57.93**±0.31 41.64±0.26 | **57.86**±0.7 42.45±0.58 | **38.63**±0.74 27.11±0.64 | **37.68**±0.73 21.28±1.26 |
| Speech Commands | 2112 | Transfer | *given* | FedorAS *rand-init* | $12 \rightarrow 35$ | **67.06**±2.12[†] 64.99±1.41[†] | **66.87**±1.85[†] 65.03±0.69[†] | **67.65**±1.59 66.55±2.2 | **68.49**±1.47 66.84±0.87 |

[†] Trained only on clients belonging to Tier 3 and 4.

standard deviation compared to the average performance of models found by `FedorAS` in Tab. 11. Additionally, we also plot detailed performance of each model trained during this process in Fig. 7 for the sake of completeness. Noticeably, `FedorAS` achieves significantly better results in almost all cases, with SpeechCommands being the only exception. We suspect this is due to problem with fine-tuning rather than architectures themselves – specifically, we witness accuracy of the discovered models can vary significantly as we repeat the fine-tuning process (this is also visible in the case of full training, although the extend is smaller). Consequently, although on average `FedorAS` performs slightly worse, in many cases the best results surpass that of random search – because of that we suspect that fine-tuning for longer would improve `FedorAS`'s performance. We leave this for future work, considering that a single shortcoming like that does not seem significant in the light of the rest of our results.

### E.2 COMPARISON AGAINST CENTRALISED NAS

To study the impact that FL has in supernet-based NAS, we train the CIFAR-10 supernet in a similar centralised setting to `FedorAS`'s. Similarly to Stage I in FedorAS, we do 500 training epochs following the SPOS paradigm but, because it is centralised, we do not impose any FLOPs budgets when sampling paths along the supernet. After training, we use NSGA-II to search for the best performing architecture/path and select the best among 1000 valid models for each tier. Finally, these models are trained from scratch for 200 epochs (again in a centralised setting, i.e., having access to all training data). The resulting test accuracies per tier are shown in the Tab. 12 and compared against the federated IID setting of CIFAR-10 (the simplest FL setup).

## F IMPACT OF WEIGHT-SHARING ON DIFFERENT TASK TRAINING

In this section, we expand on the analysis of Sec. 4.3. Specifically for CIFAR-100 and SpeechCommands, we also create a second scenario, where not all clients train in the same domain of labels. For CIFAR-100, there are 20 superclasses that are coarse-grained categories of the 100 standard labels. In contrast, for SpeechCommands, there are 12 and 35-class label sets, with the latter annotating the "other" class more specifically. In both cases, we assume a non-uniform distribution of clients to clusters, assuming that most data (75-80%) reside on lower-tier devices, whereas the two higher

**Table 14:** The effect of Stage-III of `FedorAS`: tier-aware federated finetuning of models extracted from the supernet. This table reports the increase in validation accuracy (measured in percentage points) for classification tasks of the model after finetuning and the decrease in perplexity for next word prediction (i.e. Shakespeare).

| Dataset | Tier 1 | Tier 2 | Tier 3 | Tier 4 | Tier 5 |
|---|---|---|---|---|---|
| CIFAR-10$_{\alpha=1000}$ | $1.23\pm0.11$ | $0.80\pm0.16$ | $1.18\pm0.02$ | $1.11\pm0.19$ | N/A |
| CIFAR-10$_{\alpha=1.0}$ | $3.99\pm0.22$ | $3.86\pm1.36$ | $3.55\pm0.68$ | $2.63\pm0.49$ | N/A |
| CIFAR-10$_{\alpha=0.1}$ | $8.27\pm1.34$ | $7.44\pm1.12$ | $7.78\pm1.39$ | $7.76\pm2.57$ | N/A |
| CIFAR-100$_{\alpha=0.1}$ | $8.56\pm0.04$ | $8.69\pm0.33$ | $7.20\pm0.34$ | $7.74\pm1.04$ | N/A |
| Speech Commands | $4.66\pm0.06$ | $5.68\pm0.55$ | $7.98\pm1.11$ | $6.71\pm1.32$ | N/A |
| Shakespeare | $0.273\pm0.025$ | $0.220\pm0.028$ | $0.227\pm0.046$ | $0.210\pm0.020$ | $0.190\pm0.020$ |

tier-devices (holding 20-25% of data) can train on the fine-grained label set. Our aim is to test whether few data allocation on the high tier devices can benefit from the knowledge and feature extraction learned from the coarse-grained label set. We adopt two different setups. For CIFAR-100, we train both tasks simultaneously, having essentially two distinct linear layers across tiers (1,2) and (3,4). We call this setup multi-task. On the other hand, for SpeechCommands, we train all clients on the same coarse domain and subsequently transfer learn a fine-grained 35 classes linear layer for the client of tiers (3,4). We fine-tune the linear layer keeping the rest of the network frozen for 25 epochs and then allow for finetuning of the whole network for another 100. We present the results in Tab. 13.

In both cases `FedorAS` learns better models and transfer knowledge from the low-tiers to high-tiers and vice-versa through weight sharing of the supernet. Indicatively, for CIFAR-100 we are able to achieve +14.91 pp (percentage points) of accuracy compared to training the same architectures end-to-end on eligible devices. Similarly, we achieve +1.6 pp higher accuracy for SpeechCommands.

# G  CONVERGENCE AND `FedorAS` PER-STAGE ANALYSIS

## G.1  IMPACT OF STAGE III: FEDERATED FINETUNING

After Stage I in `FedorAS`, models extracted from the supernet can directly be used for the target task (e.g. image classification). However, performance on this target task can be improved further by finetuning these models. This is the purpose of Stage III. In Table 14 we show the increase in model quality (measured on the global validation set) before and after Stage III.

## G.2  CONVERGENCE AND IMPACT OF SUPERNET SAMPLING

The proposed OPA aggregation scheme converges faster than an alternative aggregation method that does FedAvg of the updated supernets. We show this in Fig. 8. When supernets from each client participating in the round get aggregated with OPA, the resulting supernet is consistently better in terms of validation accuracy than when aggregation is done with FedAvg. This difference becomes more evident as $B_{\Phi_{comm}}$ is reduced, i.e., as smaller portions of the supernet are sent to the clients. In Fig. 8 (d) we asses the feasibility of reducing the number of rounds and end the federated supernet training stage when validation accuracy reaches approximately 70%. Those points correspond to 250 rounds and 150 rounds for the setting when 50% and 25% of the supernet is sent to the clients, respectively. The results that these settings yield show a significant loss (compared to their respective settings but over 500 rounds) and we therefore also run the same settings but when allowing for 50 additional runs. We observe a significant jump in per-tier model performance. We leave as future work investigating alternative metrics to more accurately (but without incurring into heavy computational costs) measure the quality of the supernet at any given training iteration and leverage this to better inform an early stopping mechanism to further reduce communication costs.

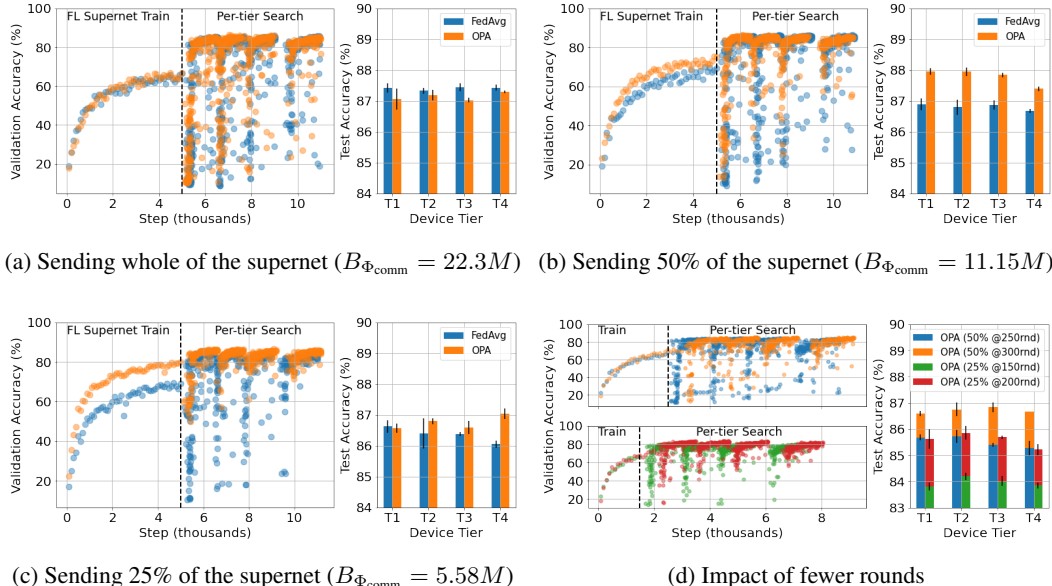

(a) Sending whole of the supernet ($B_{\Phi_{comm}} = 22.3M$)  (b) Sending 50% of the supernet ($B_{\Phi_{comm}} = 11.15M$)

(c) Sending 25% of the supernet ($B_{\Phi_{comm}} = 5.58M$)  (d) Impact of fewer rounds

**Figure 8:** Results using OPA compared to FedAvg in `FedorAS` for CIFAR-10 non-IID ($\alpha = 1.0$) with fixed hyperparameters. Each sub-plot contains two plots: first a scatter plot that visualises the federated supernet training (Sec. 3.1) in the first 5k steps and the per-tier search stage (Sec. 3.2) the remaining steps; and, a bar plot that shows the average test accuracies of each tier after federated finetuning (Sec. 3.3). For (a)-(c), the training setups are identical with the exception of using either OPA or FedAvg. Both settings perform 500 rounds of federated supernet training using 10 clients per round (i.e. 5k steps). Every 10 rounds, the supernet is evaluated on the global validation set by randomly sampling paths and a dot is added to the plot. In the context of reduced communication budget, OPA largely outperforms FedAvg, requiring fewer federated rounds to reach the same validation accuracy. This difference is more noticeable when sending 50% of the supernet (roughly corresponding to the size of a ResNet18 model). In (d) we measure the quality of the models found when the supernet training phase ends once 70% validation accuracy is reached – the highest reached by FedAvg in (a)-(c) – and show competitive performance of models derived from OPA-aggregated supernets. By extending the training phase by 50 rounds, we observe large improvements in the quality of the final models.

In Figure 9 we show the behaviour of `FedorAS` across all stages for the datasets considered in this work. These analysis uses the intermediate results generated for one of the seeds in Table 5 using the default hyperparameters as per Tables 7&8 and with all datasets using as communication budget, $B_{\Phi_{comm}}$, half of their respective supernet sizes.

# H  BEHAVIOUR UNDER ALTERNATIVE TIER CLUSTERING

## H.1  SUPERNET WITH WIDER FLOPS RANGE

We design a larger supernet for CIFAR-10 by stacking more blocks (4 blocks with 4 searchable layers followed by two blocks of 8 searchable layers) in the supernet while not requiring the reduction block between supernet blocks to always downsample the input (see D.3). This is done to prevent the width and height of activations to decrease too rapidly (i.e. after each block). The MFLOPs limits for each tier are: [101.06, 714.01, 1326.95, 4450.72], with the smallest model in Tier 1 (i.e. 53.77 MFLOPs) representing the fixed costs of the supernet (FLOPs involving non-searchable layers). With this setup, Tier 4 models can have a $82\times$ larger compute footprint than Tier 1 models. Results are depicted in Table 15. These were obtained using the exact same hyperparemeter configuration (see Table 7 and Table 8 for hyperparameters of FedorAS Stage I and Stage-II & III respectively) as the CIFAR-10 experiments shown in Table 5, with the exception of now having a larger supernet. Still, the candidate operators in the supernet remain the same and so is the proportion of clients assigned to each tier. For these experiments we followed the same procedure when establishing the communication budget and therefore we set it to be half of the supernet size. We report the results of the finetuning stage of FedorAS and show improvement over those obtained with the smaller (original) supernet. We also report the resulting performance when networks are randomly initialised (i.e. *rand-init* setup).

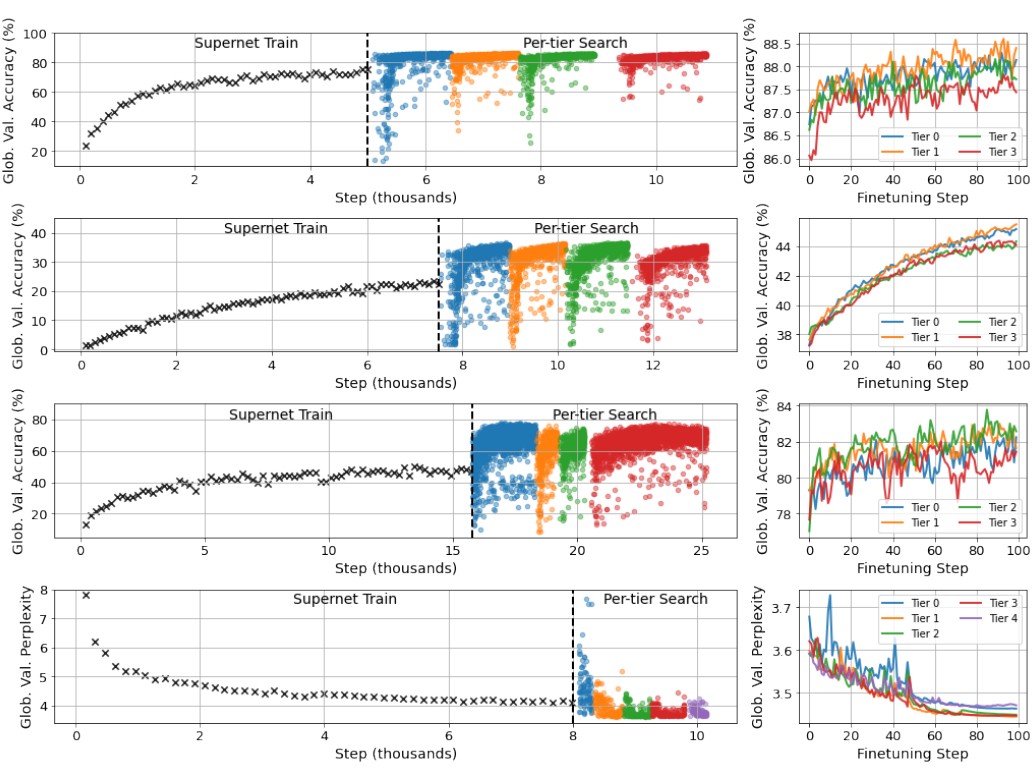

**Figure 9:** Convergence analysis for each of the datasets considered in this work (from top top bottom: CIFAR-10, CIFAR-100, Speech Commands and, Shakespear). For each dataset, the three stages in FedorAS are shown in the first subplot in the row (Stages I&II) and the subplot on the right shows the final per-tier fientuning (Stage III). Per-tier search and finetuning is colour-coded.

**Table 15:** Performance of `FedorAS` when increasing the FLOPs range of models in the supernet. Models in Tier 4 are up to 82× larger than dose in Tier 1.

| Dataset | Tier 1 | Tier 2 | Tier 3 | Tier 4 |
|---|---|---|---|---|
| CIFAR-10$_{\alpha=1.0}$ | 89.54±0.12 (+3.55pp) | 89.89±0.10 (+3.59pp) | 89.99±0.19 (+3.65pp) | 89.34±0.41 (+2.88pp) |
| CIFAR-10$_{\alpha=0.1}$ | 84.42±0.41 (+3.41pp) | 84.94±0.42 (+3.41pp) | 84.11±1.37 (+3.47pp) | 83.63±1.13 (+2.78pp) |

**Table 16:** Performance of `FedorAS` when scaled to 8 tiers.

| Dataset | Tier 1 | Tier 2 | Tier 3 | Tier 4 | Tier 5 | Tier 6 | Tier 7 | Tier 8 |
|---|---|---|---|---|---|---|---|---|
| CIFAR-10$_{\alpha=1.0}$ | 87.53±0.29 | 87.81±0.21 | 87.59±0.57 | 87.84±0.26 | 87.65±0.20 | 87.76±0.40 | 87.21±0.55 | 87.11±0.45 |
| CIFAR-10$_{\alpha=0.1}$ | 80.34±1.70 | 81.56±0.46 | 81.50±1.19 | 81.54±0.69 | 81.12±0.81 | 81.14±0.29 | 81.28±1.13 | 79.92±1.94 |

The results in Table 15 tell us that even in the scenario where the span of model footprints is larger (82× in this experiment): small models (Tier 1 – up to 101 MFLOPs) trained with FedorAS can very well take advantage of larger models trained on other more capable clients (Tier 2 and above); and, better models are obtained across Tiers showing large gains compared to the original supernet (considering models up to 700MFLOPs approximately).

## H.2 SCALABILITY TO MORE CLUSTERS

So far, we have shown results on a set number of clusters. In this section, we scale up the number of clusters from 4 to 8 and show the behaviour of our system in the case of CIFAR-10$_{\alpha=\{1.0,0.1\}}$. Results are depicted in Tab. 16.

## H.3 DIFFERENT ALLOCATION OF CLIENTS TO TIERS

In this experiment, contrary to what was described in Appendix D.4, we adopt the device capabilities trace from Oort [44] for our device to cluster allocation and performed training on CIFAR-10 and CIFAR-100. Results are depicted on Tab. 17. It can be witnessed that `FedorAS` is able to operate even under harsher heterogeneity conditions and still output competitive models in the federated setting.

## I FAIRNESS

Up to this point, we have reported on the global test accuracy, which is a union over the test sets of the clients for all datasets but SpeechCommands, which defines a separate set of clients mutually exclusive to the ones participating in federated training. In this section we want to measure how the ability of our system to aggregate knowledge across tiers during supernet training affects the fairness in participation. We quantify this through variance and worst-case statistical measures of client performance on their respective test set. These test sets are considered to be following the same distribution (LDA or pre-split) of their train and validation sets.

Results are shown on Tab. 18 for the different datasets offering per-client test data splits for all clients. It can be seen that `FedorAS` leads consistently to better performance compared to end-to-end FL trained networks of the same architectures, where only eligible clients can train models. With respect to the standard deviation of per client performance, we witness `FedorAS` offering lower variance, except for the case of CIFAR-100. We consider this to be a consequence of our significantly higher

**Table 17:** `FedorAS` performance under [44] device clustering. We see that performance still scales well, albeit taking an impact due to more extreme system heterogeneity compared to results from Tab. 5. This hints that weight-sharing through our supernet works well under varying device allocation settings.

| Dataset | #clients | Method | Perf. |
|---|---|---|---|
| CIFAR10$_{\alpha=1}$ | 100 (10) | `FedorAS`$_{\text{per tier}}$ | [**87.77**±**0.09**, **87.43**±**0.28**, **87.15**±**0.42**, **87.01**±**0.11**] |
| | 100 (10) | *rand-init* | [85.83±1.24, 83.91±0.98, 83.21±0.60, 79.10±1.63] |
| CIFAR100$_{\alpha=0.1}$ | 500 (10) | `FedorAS`$_{\text{per tier}}$ | [**44.33**±**0.81**, **43.83**±**0.89**, **43.72**±**0.82**, **43.22**±**1.02**] |
| | 500 (10) | *rand-init* | [34.88±0.28, 33.76±2.53, 32.72±0.68, 30.77±2.30] |

**Table 18:** Quantification of fairness with per client accuracy statistics, comparing end-to-end with `FedorAS`'s performance. We report on the mean, standard deviation per tier and across tiers and minimum performance (min accuracy or max perplexity) across datasets with per-tier client test sets. Lower standard deviation is better.

| Dataset | Mode | # Total Clients | Perf. (across tiers) | Perf. (per tier) | Worst Perf. |
|---|---|---|---|---|---|
| CIFAR-10 (Acc. (%) ↑ is better) | Fedoras *rand-init* | 100 | **81.36**±**8.58** 65.70 ±13.11 | [**82.43**±**7.87**, **81.41**±**9.34**, **81.78**±**8.49**, 79.80±**8.42**] [67.43±12.29, 66.10±14.05, 66.14±13.02, 63.12±12.76] | **47.00** 11.00 |
| CIFAR-100 (Acc. (%) ↑ is better) | Fedoras *rand-init* | 500 | **45.61**±**13.15** 31.08±**12.12** | [**44.46**±13.81, **45.51**±12.28, **45.19**±13.15, **47.29**±13.15] [30.72±**13.16**, 30.65±**11.27**, 30.43±**11.25**, 32.52±**12.59**] | **5.00** 0.00 |
| Shakespeare (Perplexity ↓ is better) | Fedoras *rand-init* | 715 | **2.93**±**1.01** 3.07±1.10 | [**2.93**±**0.97**, **2.94**±**0.99**, **2.88**±1.03, **2.98**±1.04, **2.94**±**1.01**] [3.07±1.05, 3.08±1.10, 3.02±1.13, 3.11±1.13, 3.07±1.10] | 8.43 **8.36** |

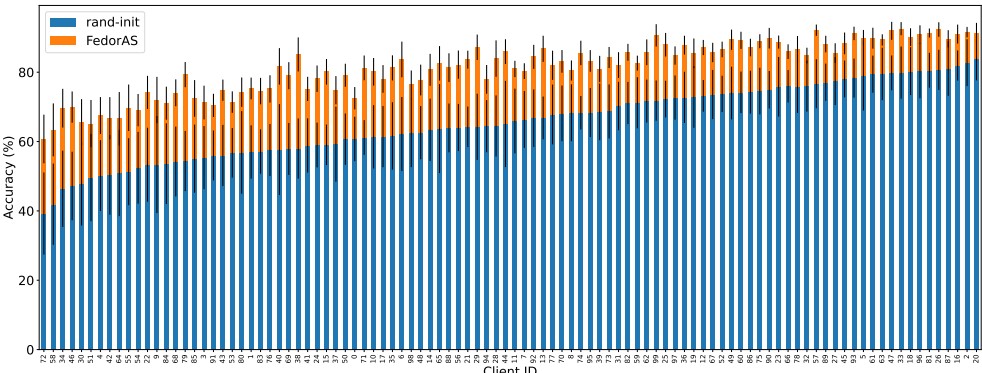

**Figure 10:** Accuracy per client of `FedorAS` vs. end-to-end (rand-init) FL-trained models (same architecture) on CIFAR-10. Accuracy is quantified on each client's dataset from the model associated with that device's tier and error bars represent the standard deviation of accuracy between different runs. Across runs, the allocation of data to clients does not change. Ordered by ascending end-to-end accuracy.

accuracy. Similar behaviour, is also witnessed when we measure variance per tier of devices. Last, we also showcase the worst-case result of a clients performance in the last column of the table.

An in-depth view of how each client behaves is depicted in Fig. 10 for CIFAR-10, where we show performance of Fedoras vs. end-to-end FL-trained models per client.

## J    EVALUATION OF THE SEARCH PHASE

### J.1    SENSITIVITY TO SIZE OF VALIDATION SET

In order to more meaningfully examine the impact of the validation set size, we run experiments on all datasets with 20% and 50% of the size of the initial global validation set. This means that during Stage-II of `FedorAS`, architectures sampled from the supernet are scored using a fraction of the global validation set. These validation subsets are extracted uniformly. After obtaining the best performing models for each tier, these are fine-tuned in a federated fashion (Stage-III in `FedorAS`) for 100 rounds just like it was done for Tab. 5. We maintain the same hyperparameters as those used to generate Tab. 5. Results are depicted in Tab 19 for all datasets.

### J.2    FEDERATED SEARCH

In this Section, we provide additional details and results concerning our federated variant of NSGA-II, discussed in Sec. 4.5 of the main paper. Below, we provide some context about how the algorithm works, how we setup the federated experiments and commentary of the results on CIFAR-$10_{\{1000,1,0.1\}}$ and CIFAR-100.

**Table 19:** Test accuracy for different sample size of validation set. Results are shown as relative change in final test accuracy for each tier compared to the scenario where the whole validation set is used. Results for each dataset are averaged over three runs.

| Dataset | Clients | Partitioning | Val set prct. | Classes | Tier 1 | Tier 2 | Tier 3 | Tier 4 |
|---|---|---|---|---|---|---|---|---|
| CIFAR-10 | 100 | $IID_{\alpha=1000}$ | 1.0 | 10 | $89.40_{\pm0.19}$ | $89.60_{\pm0.15}$ | $89.64_{\pm0.22}$ | $89.24_{\pm0.29}$ |
| | | | 0.5 | | -0.31 | -0.10 | -0.14 | **+0.03** |
| | | | 0.2 | | -0.06 | -0.17 | -0.19 | -0.02 |
| | | non-IID$_{\alpha=1.0}$ | 1.0 | 10 | $85.99_{\pm0.13}$ | $86.30_{\pm0.41}$ | $86.34_{\pm0.19}$ | $85.58_{\pm0.55}$ |
| | | | 0.5 | | **+0.06** | -0.18 | **+0.19** | -0.19 |
| | | | 0.2 | | -0.05 | +0.02 | -0.28 | -0.29 |
| | | non-IID$_{\alpha=0.1}$ | 1.0 | 10 | $81.01_{\pm0.46}$ | $81.53_{\pm0.29}$ | $80.64_{\pm0.66}$ | $80.85_{\pm0.28}$ |
| | | | 0.5 | | **+0.63** | -0.5 | -0.29 | -0.14 |
| | | | 0.2 | | **+0.78** | **+0.13** | -1.55 | -0.09 |
| CIFAR-100 | 500 | non-IID$_{\alpha=0.1}$ | 1.0 | 100 | $45.25_{\pm0.13}$ | $45.84_{\pm0.18}$ | $45.42_{\pm0.39}$ | $45.07_{\pm0.71}$ |
| | | | 0.5 | | -0.10 | **+0.01** | -0.27 | -0.38 |
| | | | 0.2 | | **+0.07** | **+0.11** | **+0.32** | -0.60 |
| Speech Commands | 2112 | *given* | 1.0 | 12 | $80.19_{\pm1.78}$ | $80.47_{\pm1.69}$ | $81.0_{\pm1.58}$ | $80.56_{\pm0.40}$ |
| | | | 0.5 | | **+0.40** | **+0.64** | **+0.25** | **+1.76** |
| | | | 0.2 | | -0.98 | -0.40 | -1.49 | **+1.08** |

| Dataset | Clients | Partitioning | Val set prct. | Classes | Tier 1 | Tier 2 | Tier 3 | Tier 4 | Tier 5 |
|---|---|---|---|---|---|---|---|---|---|
| Shakespeare | 715 | *given* | 1.0 | 90 | $3.43_{\pm0.01}$ | $3.39_{\pm0.04}$ | $3.38_{\pm0.03}$ | $3.40_{\pm0.01}$ | $3.42_{\pm0.01}$ |
| | | | 0.5 | | -0.004 | -0.000 | -0.014 | -0.015 | -0.004 |
| | | | 0.2 | | -0.002 | -0.07 | -0.003 | **+0.003** | -0.008 |

**Algorithm details.** Federation is achieved by delegating evaluation of models to individual clients. This means that evaluation is done stochastically on clients' local datasets. This setting is similar to Federated Evaluation in [64]. In order to avoid sending a large number of models to clients, thus saving communication cost, we leverage the fact that NSGA-II operates in "batches" of models – in each iteration, a number of models (i.e. *sample size*) from the *population* is selected to produce new models that replace the ones not selected. Since models from a single "batch" are all selected at the same time, we do not need to evaluate them sequentially. Instead, considering that weights between architectures are shared, we can construct a minimal supernet that encapsulates all selected models and send it to relevant clients. This way, we can achieve parallel evaluation of all selected models and optimal[4] communication cost for a given "batch".

**Experimental setting.** The experimental setting is following exactly what was used in other experiments. Specifically, population and sample sizes were the same as the ones used in centralised NSGA-II (128 and 64, respectively), and federated evaluation (FE) rounds were analogous to rounds during federated training of the supernet from which models to evaluate were extracted. Concretely, number of clients, allocation of clients to tiers, allocation of data to clients, number of available clients per round, and client selection mechanism were all exactly the same for FE as the ones used during the relevant Stage 1 of `FedorAS`.

However, to present more meaningful results, we assumed that as we fluctuate the number of clients used to evaluate models (number of evaluation rounds) additional clients are always unique. In other words, reshuffling and "forgetting" of clients is happening only between federated NSGA-II iterations. Consequently, it is possible that: *i)* if all clients are used to evaluate a model (e.g. 10 evaluation rounds for CIFAR-10), FE is equivalent to centralised evaluation; *ii)* with a relatively small number of FE rounds, it is possible that one set of models produced by NSGA-II is evaluated on a completely disjoint set of data to another "batch"; *iii)* similarly, we do not enforce in any way that all clients have to be used (unless we use all clients in a single iteration of NSGA-II). Finally, keeping consistent with the rest of the paper, we performed evaluation in a tier-aware manner, meaning that models belonging to higher tiers could only access a subset of all validation data based on client eligibility. This is the reason why relevant curves finish at a different maximum number of FE rounds, that is a smaller number of FE rounds (i.e. Federated NSGA-II iterations) is needed to exhaust the set of eligible clients.

**Results.** Fig. 11 complements Fig. 4 and presents results across three different experimental settings, with varying level of non-IID-ness ($\alpha \in \{1, 0.1\}$) and number of clients ($\{100, 500\}$). As conjectured in the main paper, we can observe that the cost of achieving the same fidelity of FE increases with both the level of non-IID-ness and the number of clients. Specifically, Kendall-$\tau$ of 0.8 is achieved

---

[4]Optimal in the sense of communicating paths once and not accounting for orthogonal techniques such as compression, etc.

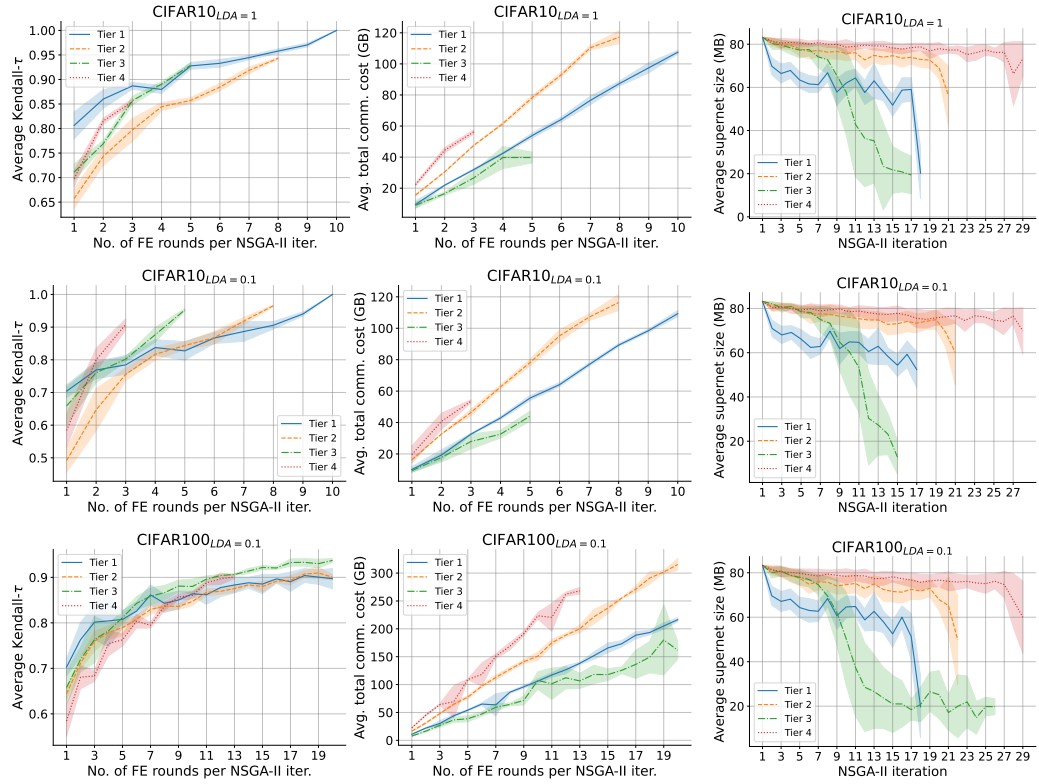

**Figure 11:** Ranking quality & cost of federated evaluation (FE) of models during federated search. Each time a new population of models is evaluated, a minimal supernet encompassing selected models is sent to a sample of clients; **left)** ranking correlation between scores produced by FE & centralised evaluation (CE), as a function of FE rounds (↑ rounds = ↑ clients); **middle)** total communication cost of sending all necessary supernets to all clients, to run a full search; **right)** changes in the supernet size as NSGA-II progresses.

at approximately three FE rounds for $\{\alpha = 1, \text{clients} = 100\}$, and increases to four and six for $\{\alpha = 0.1, \text{clients} = 100\}$ and $\{\alpha = 0.1, \text{clients} = 500\}$, respectively. At the same time, we can see that regardless of the setting NSGA-II tends to produce smaller supernets as the search progresses, suggesting that searching for longer does not have to incur proportional increase in the communication cost.

## J.3 CORRELATION WITH POST-FINE TUNING ACCURACY

The main objective of the NSGA-II search in Stage 2 is to find the most accurate models, where accuracy is understood as validation accuracy immediately after extracting a relevant subnet from a pretrained supernet. The assumption is that the better a model performs in a case like that, the better it will be after the following fine tuning performed in Stage 3. However, weight-sharing NAS is well-known to fail to meet this assumption in many cases [82, 83, 85, 78, 61]. Therefore, we additionally quantify the fidelity of out searching objective by measuring ranking correlation between validation accuracy of 160 random models from a search space, when taken directly from a supernet after Stage 2, to their final test accuracy after fine tuning in Stage 3. Results are presneted in Figure 12.

We can see that in general ranking models based on their accuracy when directly using weights from a supernet is not very faithful to how good a model can be with extra fine-tuning, which is aligned with the aforementioned observations made in the centralised setting. While this shows that further improvements could be achieved with more work on the searching algorithm, we would argue that to obtain more indicative performance of models, one would need to perform additional training, which in our settings involves following the entire FL procedure, making the cost of doing so even more non-negligible. On the other hand, while the best models identified after Stage 2 are unlikely to be

**Figure 12:** Correlation between validation accuracy as measured during Stage 2 and the final test accuracy of a model after fine-tuning in Stage 3, for 160 models randomly selected from a relevant search space. **Left)** CIFAR-10$_{\alpha=1.0}$, **middle)** CIFAR-10$_{\alpha=0.1}$, **right)** CIFAR-100$_{\alpha=0.1}$. Note: unlike the rest of the paper, CIFAR-10 supernets were trained for 750 epochs here, not 500.

the best after fine-tuning, they tend to lean towards better performing ones. Therefore, based on our strong empirical results and observed rankings in Figure 12, we would conclude that they constitute a safe choice which, at the same time, involves minimal searching cost. Due to those reasons, we leave the challenge of improving the fidelity of validation scores for future work.

