# OpenReview forum: "FedorAS: Federated Architecture Search under system heterogeneity"
_ICLR.cc/2023/Conference — Submitted to ICLR 2023_

### Official Review · Reviewer_Zb8v · 2022-10-24

**Confidence:** 3
**Correctness:** 3
**Technical Novelty And Significance:** 2
**Empirical Novelty And Significance:** 2
**Recommendation:** 6

**Clarity, Quality, Novelty And Reproducibility:**

Overall quality and clarity of the paper are good. There are some rooms for improving the originality of the proposed method. For example, discussion of the clear difference from some existing methods that utilizes the supernet for Federated NAS (i.e. HeteroFL). The code is not provided, thus it is hard to tell the reproducibility of this work.


**Strength And Weaknesses:**

**Strength**
- The paper is well-organized and easy to read.
- They tackle the practical problem of federated learning where participants have device heterogeneity, i.e. computation and memory budgets.
- They demonstrate the effectiveness of the proposed method in various modalities, i.e. vision, speech, text.
- The results are impressive.

**Weaknesses**
- It seems better to discuss the overall costs (i.e. supernet train and model search time (ms) or # rounds, etc) and compare them to the baseline models. Since the proposed framework has three steps; (1) supernet training, (2) model search, and (3) local fine-tuning, each step may take the individual computation and communication costs. In the paper, however, I can only find the analysis for the searched models obtained from the step 3 only. More detailed discussion for overall costs is required, for example:
    - How many rounds do the stage (1) and stage (3) need?
    - How long does the (2) model search process take?
    - How much overall computation and memory costs are required to get the final results for your model and baseline models?
- Is the number of “tiers” the hyper-parameters? Can we have more tiers? Does varying the number of tiers affect the performance? or the overall costs for each step?
- HeteroFL [Diao et al 21], one of the essential works tackling the same problem, is missing. The method also leverages the supernet for sampling local models considering the complexity levels, which is also similar to your tier clustering. It would be very helpful for you to discuss pros and cons and compare it with the proposed model.

Diao et al, HeteroFL: Computation and Communication Efficient Federated Learning for Heterogeneous Clients, ICLR 2021


**Summary Of The Paper:**

The authors propose a method for federated neural architecture search, namely FedorAS, which leverages supernet for sharing knowledge across clients with device and data heterogeneity. They also introduce OPA (operation aggregation) that weighs the client updates in a frequency-aware manner. They demonstrate the effectiveness of their method against multiple existing methods.

**Summary Of The Review:**

I enjoyed reading the paper, but several improvement seems to be required, as mentioned above.

---

> ### Author Response · Authors · 2022-11-17
> **Replying to Reviewer's Questions (1/2)**
>
> To make this reply self-contained we first indicate the question raised and then add our response.
>
> > _It seems better to discuss the overall costs (i.e. supernet train and model search time (ms) or # rounds, etc) and compare them to the baseline models. Since the proposed framework has three steps; (1) supernet training, (2) model search, and (3) local fine-tuning, each step may take the individual computation and communication costs. In the paper, however, I can only find the analysis for the searched models obtained from the step 3 only. More detailed discussion for overall costs is required, for example:_
> >
> > *   _How many rounds do the stage (1) and stage (3) need?_
>
> Number of rounds for each experimental setting can be found in Appendix D.5, Tables 7-9. Specifically, stage 1 takes 500-750 rounds, and stage 3 takes 100 rounds per tier.
>
> > * _How long does the (2) model search process take?_
>
> We search for 100-1000 valid models per each tier, depending on a search space size (2000 for speech commands is only used in one of the additional experiments), using population size of 128, and sample size of 64. Table 8 summarises this information per dataset. Depending on the validation method (centralised, centralised sampled, federated from Sec. 4.5), this step can vary in duration. Under all cases, however, this was never a bottleneck in the overall process.
>
> > * _How much overall computation and memory costs are required to get the final results for your model and baseline models?_
>
> The computation costs of federated supernet training, the compute costs of the finetuning stage and memory cost in terms of memory peak (minimum memory that a device needs to run any possible model) in cross-device setups are shown in the table below. The memory peak is a function of the model architecture, the input size and the batch size. Note that the compute footprints account for the total costs (across rounds, clients and local epochs) for both supernet training and finetuning stages.
>
> The performance column shows the average performance of each method for a particular alpha. If the method is tier-aware (i.e. FedorAS, FjORD) we report the average performance across tiers. Note that the compute cost of supernet training (stage 1 in FedorAS) can be further amortised by subsequent finetuning stages when considering more tiers.
>
> Overall, FedorAS stands as a more versatile FL+NAS solution than FedNAS given that FedorAS is tier aware (the memory peak of the smallest tier is ~175MB) and delivers more consistent results at different alphas. Recall that if the memory peak of FedNAS is lowered to match that of FedorAS, its performance gets severely hurt (as already shown in Table 2). Compared to FjORD (the alternative tier-aware solution), FedorAS delivers substantially better results (CIFAR-10/Shakespeare in Table 4)  while keeping the inherent costs to NAS+FL (i.e. larger memory peaks, higher compute footprints) under control and respecting the capabilities of each device tier. For SpeechCommands, FedorAS is orders of magnitude more compute efficient than Oort and PyramidFL and with 4x lower memory peak while still offering a large improvement in accuracy.
>
> | Experiment|Method|Total Compute on Clients for Supernet training (GFLOPs)|Finetune Compute Costs (GFLOPs)|Memory Peak (MB) of largest model encountered|Tier-aware|Performance(average for each alpha considered. Average across tiers if tier-aware experiment)|
> |:------------- |:-------------:|:-------------:|:-------------:|:-------------:|:-------------:|:-------------:|
> |Table 2|FedNAS|72 x 10^5|>0 (but unclear how much)|3837|No|[90.02, 65.28]|
> |Table 2|FedorAS|172 x 10^5|41 x 10^3|1996|Yes|[86.46, 81.53]|
> |Table 4 – CIFAR-10|ZeroFL|14 x 10^5|N/A|148|No|[82.82, 81.04, N/A]|
> |Table 4 – CIFAR-10 |FjORD|8 x 10^5|N/A|148|Yes|[78.07, 78.45, 59.85]|
> |Table 4 – CIFAR-10 |FedorAS|172 x 10^5|41 x 10^3|1996|Yes|[89.47, 86.27, 81.01]|
> |Table 4 – Shakespeare|FjORD|2 x 10^3|N/A|3.2|Yes|3.99|
> |Table 4 – Shakespeare|FedorAS|29 x 10^3|767|5.9|Yes|3.39|
> |Table 4 – Speech Commands|Oort|477 x 10^4|N/A|232|No|62.20|
> |Table 4 – Speech Commands|PyramidFL|239 x 10^4|N/A|248|No|63.84|
> |Table 4 – Speech Commands|FedorAS|5 x 10^4|312|54|Yes|67.53 (from table 13)

---

> > ### Author Response · Authors · 2022-11-17
> > **Replying to Reviewer's Questions (2/2)**
> >
> > > _Is the number of “tiers” the hyper-parameters? Can we have more tiers? Does varying the number of tiers affect the performance? or the overall costs for each step?_
> >
> > It is a hyperparameter and can be freely changed. The cost of stage 1 stays the same regardless of the number of tiers. The cost of stages 2 and 3 scales linearly. Regarding performance, see our reply to reviewer 7XQj.
> >
> > > _HeteroFL [Diao et al 21], one of the essential works tackling the same problem, is missing. The method also leverages the supernet for sampling local models considering the complexity levels, which is also similar to your tier clustering. It would be very helpful for you to discuss pros and cons and compare it with the proposed model._
> >
> > We thank the reviewer for pointing the work of HeteroFL out. We would group this line of work along with FjORD under the dynamic width scaling techniques for system heterogeneity. Both works start with a global model architecture (the "supernet" as reviewer suggests) and then vary the width of the model's channels (or neurons) to come up with smaller subnetworks. Subnetworks are then sent to clients of different capabilities to train. While their main goal is seemingly the same, they differ in their implementation details (e.g. Scaling, Batch Normalisation, Loss function, ordering guarantees), as well as their evaluation.
> >
> > In contrast to these lines of work, which select uniformly a keep rate for each layer's channel width, FedorAS offers more degrees of freedom for matching a target footprint by offering different operator choices per layer.
> > Specifically, in both cases (HeteroFL and FjORD) the channel keep rates are uniform across layers, and the search is essentially one-dimensional. This is not the case with FedorAS, which enables each path to make independent decisions about the operators of which it consists. This creates a larger search space, which is traversed through NAS, and also allows for higher flexibility in the architectures and more fine-grained weight-sharing (operator vs. width level).
> >
> > On the other hand, FjORD (and potentially HeteroFL) creates an ordered representation in their "supernet" and enables the dynamic extraction of candidate submodels without the need to retrain or finetune. While this is possible after Stage 1 of FedorAS, we still need to finetune the best performing architectures for competitive results.
> >
> > We have incorporated a short reference in the related work due to the page limit.

---

> ### Comment · Reviewer_Zb8v · 2022-12-08
> **Thank you for the detailed responses.**
>
> Most of my concerns have been convinced except for the below:
>
> Even though FedorAS shows consistent and/or better performance against base models (throughout tiers), the computational costs and communication rounds that FedorAS requires for achieving those results seem to be relatively huge compared to the existing methods depending on the dataset.
>
> Can you explicitly compare the total requiring FLOPs (for the supernet training, fine-tuning, etc) and the total number of communication rounds for FedorAS and all baseline models? so that the overall efficiency (both resource and communication costs) can be also clearly analyzed. Indeed, I was not able to find the number of communication rounds for the baseline models in Appendix D.5, Tables 7-9.
>
> I would like to raise my score if my remaining concerns are resolved clearly.

---

> > ### Author Response · Authors · 2022-12-08
> > **Further details about communication cost**
> >
> > Thank you for the follow up - indeed there is no information about the communication cost of methods other than FedorAS, please find a summary in a table below (we analyse the cost of supernet training and subsequent fine-tuning separately for a more comprehensive overview).
> > As for the computation cost, we would expect the Table from our previous reply to be already sufficient - please let us know if that's not the case and, if so, what would need to be added.
> > We also understand that the reviewer might be referring to the fact that in the revised paper there is no mention of the cost at the moment - we will make sure it is adequately incorporated in the camera-ready and that it is clear that the benefits of using fedoras come at a certain cost.
> > However, please note that the amount of information is rather large, so most likely the details will have to go to the Appendix with appropriate f/w references in the main text.
> > Specifically, we plan to include a summary column in Table 4 together with a short comment (similar to "Correlation with final accuracy") pointing to the Appendix for details.
> >
> > **Table: Communication cost - supernet/shared model training**
> > | Experiment|Method|Supernet FL Rounds|Supernet size (M) | Num clients| Comms Costs (GB) |Notes|
> > |:------------- |:-------------:|:-------------:|:-------------:|:-------------:|:-------------:|:-------------:|
> > |Table 2 -- CIFAR-10|FedNAS|500|1.93|10|75.39| |
> > |Table 4-- CIFAR-10|FjORD|500|5.23|10|204.30|Supernet size is size of avg. model communicated to clients|
> > |Table 2&4 -- CIFAR-10|FedorAS|500|11.5|10|449.22||
> > |Table 4 -- Shakespeare|FjORD|500|0.10|10|3.83|Supernet size is size of avg. model communicated to clients|
> > |Table 4 -- Shakespeare|FedorAS|500|0.83|16|51.88||
> > |Table 4 -- SpeechCommands|FedorAS|750|1.5|21|184.57||
> >
> >
> > **Table: Communication cost - fine-tuning/training a model for a device tier** (note: methods that are not tier-aware are included here for a single tier)
> > | Experiment|Method|Standard FL Rounds|Avg model size (M) | Num clients| Comms Costs (GB) |Notes|
> > |:------------- |:-------------:|:-------------:|:-------------:|:-------------:|:-------------:|:-------------:|
> > |Table 2 -- CIFAR-10|FedNAS|?|?|?|?| Unclear how much finetuning they do after NAS|
> > |Table 2-- CIFAR-10|FedorAS|100|2.6|6|12.19|avg. per tier|
> > |Table 4-- CIFAR-10|ZeroFL|500|11.7|10|436.33||
> > |Table 4-- CIFAR-10|FjORD|0|0|0|0|FjORD does not consider fine-tuning|
> > |Table 4-- CIFAR-10|FedorAS|100|3.34|6|15.64|avg. per tier|
> > |Table 4 -- Shakespeare|FjORD|0|0|0|0|FjORD does not consider fine-tuning|
> > |Table 4 -- Shakespeare|FedorAS|100|0.20|16|2.46|avg. per tier|
> > |Table 4 -- SpeechCommands|Oort|400|21.29|1300/100|46571.88| Oort asks 1300 clients to do training but only the first 100 to finish are aggregated (we therefore only include 100 models for uplink communication) |
> > |Table 4 -- SpeechCommands|PyramidFL|400|21.29|50|3326.56| |
> > |Table 4 -- SpeechCommands|FedorAS|100|0.42|21|6.96|avg. per tier|
> >
> >
> > As can be seen, the majority of the cost is again coming from the supernet training stage with fine-tuning remaining rather lightweight, similar to computation cost.
> > This is a typical characteristic of many NAS systems, and notably supernet training can, in general, be amortised if there is a need to fine-tune more models (e.g., like in OFA [1]).
> > At the same time we can see that the overall cost of FedorAS is not prohibitively large - depending on their exact setting, some existing works, like Oort or Pyramid FL, might require significantly more compute and communication, while others might be more efficient.
> > Interestingly, while FedNAS uses much smaller supernets, keeping comms cost lower, their on-device memory usage is much higher (see, e.g. Table 2) - this is related to the (significant) differences between theirs and ours search space and supernet training methodology.
> > We keep comparison to FedNAS with aligned search spaces for future work.
> > Also, we would like to note that although we consider fine-tuning to be an integral part of our system (for obtaining the best results), and consequently we report the cost apropriately, even without fine-tuning our models can often achieve better results than baselines.
> > For example, on CIFAR-10(alpha=0.1), where benefits of fine-tuning are the largest (see reply to reviewer: djQ3), if we decided to skip fine-tuning to save on comms and compute cost, we would still improve upon FjORD across tiers by: [11.31, 13.28, 13.14, 15.65] pp, down from [19.58, 20.72, 20.92, 23.41].
> > This also holds for Shakespeare and CIFAR-10 with different alphas.
> >
> >
> > [1] H. Cai et al. "Once for All: Train One Network and Specialize it for Efficient Deployment" ICLR'20

---

> > > ### Comment · Reviewer_Zb8v · 2022-12-11
> > > **Thank you for your response.**
> > >
> > > Thanks to the comprehensive comparison, I was able to analyze pros and cons of the proposed method against base models more clearly. I agree with:
> > >
> > > > we will make sure it is adequately incorporated in the camera-ready and that it is clear that the benefits of using fedoras come at a certain cost.
> > >
> > > Now, my doubts have been thoroughly resolved. I have raised my score accordingly.

---

### Official Review · Reviewer_djQ3 · 2022-11-02

**Confidence:** 4
**Correctness:** 4
**Technical Novelty And Significance:** 3
**Empirical Novelty And Significance:** 3
**Recommendation:** 6

**Clarity, Quality, Novelty And Reproducibility:**

The paper is easy to read and clearly written. The method proposed in the paper is novel, but the authors do not provide any code for it, which significantly undermines the reproducibility of the paper.

**Strength And Weaknesses:**

**Strengths:**
* FedorAS achieves superior performance over other baselines in cross-silo settings.

**Weaknesses:**
* An ablation study on the effect of the fine-tuning process is lacking. How much does the fine-tuning process affect the final performance?

* Experiments on a more extreme division of the device tiers are lacking. For example, in the CIFAR-10 experiments, it is assumed that the lowest tier devices are capable of 120.9MFLOPs, whereas the highest tier devices are assumed to be capable of 716.0MFLOPs. That is only about 6x difference. However, the FLOPs difference between real embedded devices and high-performance devices is more extreme, for example, the FLOPs difference between the M1 chip and the Raspberry Pi Zero is about 500x, given the average power for each device. [1]

[1] https://web.eece.maine.edu/~vweaver/group/green_machines.html

**Summary Of The Paper:**

This paper introduces a method called "FedorAS" which incorporates ideas from Neural Architecture Search into the domain of Federated Learning. In the scenario where the devices participating in a Federated Learning session has varying capabilities, and heterogeneous data distribution across devices, the goal of FedorAS is to provide each device with an efficiency-tailored architecture and parameters.

In FedorAS, the participating devices are clustered into several tiers, according to their hardware capability. Using a supernet architecture, from which a subspace is sampled for each device. The client, after receiving the subspace, trains the models within the subspace by randomly sampling the subpaths. The server then aggregates the updated client weights using a weighted averaging technique which normalizes for the number of examples passed through each subpath. After training, the server performs the NSGA-II algorithm to find the best architecture for each device tier. After that, the obtained models are fine-tuned with FedAvg among devices with the same tier or above.

The authors report that FedorAS achieves better performance than other cross-device baselines, on multiple domains such as speech, vision, and text.

**Summary Of The Review:**

FedorAS is a novel method for performing NAS in a federated learning setting, which outperforms similar existing methods. It is well-written and easy to understand, but some of the experiments are considered somewhat unrealistic, and analyses are lacking. The authors do not provide any code for reproducibility.

---

> ### Author Response · Authors · 2022-11-17
> **Replying to Reviewer's Questions**
>
> To make this reply self-contained we first indicate the question raised and then add our response.
>
> > _An ablation study on the effect of the fine-tuning process is lacking. How much does the fine-tuning process affect the final performance?_
>
> In the current manuscript, one could find a subset of this information for CIFAR-10$_{α=1.0}$ on Figure 8 of the Appendix. However, for the sake of completeness we have assembled a table summarising this information across datasets. The table below shows the effect of fine-tuning (stage III in FedorAS) the models extracted from the supernet (stage II in FedorAS) and their performance measured on the global validation set. We report these metrics in terms accuracy increase in percentage points (pp) for classification tasks (i.e. CIFAR10/100 and SpeechCommands) and decrease in perplexity (p) for next word prediction tasks (i.e. Shakespeare) between Stages 1 and 3. The results are extracted from the experiments that resulted in Table 5.
>
> | Dataset | Tier 1 | Tier 2 | Tier 3 | Tier 4 |  Tier 5 |
> |:------------- |:-------------:|:-------------:|:-------------:|:-------------:|:-------------:|
> |CIFAR-10 (α=1000)|1.23±0.11|0.8±0.16|1.18±0.02|1.11±0.19|n/a|
> |CIFAR-10 (α=1)|3.99±0.22|3.86±1.36|3.55±0.68|2.63±0.49|n/a|
> |CIFAR-10 (α=0.1)|8.27±1.34|7.44±1.12|7.78±1.39|7.76±2.57|n/a|
> |CIFAR-100 (α=0.1)|8.56±0.04|8.69±0.33|7.20±0.34|7.74±1.04|n/a|
> |Speech Commands|4.66±0.06|5.68±0.55|7.98±1.11|6.71±1.32|n/a|
> |Shakespeare|0.273±0.025|0.220±0.028|0.227±0.046|0.210±0.020|0.190±0.020|
>
> > _Experiments on a more extreme division of the device tiers are lacking. For example, in the CIFAR-10 experiments, it is assumed that the lowest tier devices are capable of 120.9MFLOPs, whereas the highest tier devices are assumed to be capable of 716.0MFLOPs. That is only about 6x difference. However, the FLOPs difference between real embedded devices and high-performance devices is more extreme, for example, the FLOPs difference between the M1 chip and the Raspberry Pi Zero is about 500x, given the average power for each device. [1]_
>
> We acknowledge this is a current limitation of our evaluation, which is a result of our supernet definition. To remedy this, we have run an extra experiment on CIFAR-10 alpha=[0.1,1.0] where we tweak the supernet to encompass networks with FLOPs in the range of [53.8M - 4450M] FLOPs. The results are shown in the table below, with large increases in classification accuracy across all tiers.
>
> These results tell us that even in the scenario where the span of model footprints is larger (`84x` in this experiment): small models (Tier 1 – up to 101 MFLOPs) trained with FedorAS can very well take advantage of larger models trained on other more capable clients (Tier 2 and above); and, better models are obtained across Tiers showing large gains compared to the original supernet (considering models up to ~700MFLOPs).
>
> | Dataset | Tier 1 | Tier 2 | Tier 3 | Tier 4 |
> |:------------- |:-------------:|:-------------:|:-------------:|:-------------:|
> |CIFAR-10 (α=1.0)|89.54±0.12 (+3.55)|89.89±0.10 (+3.59)|89.99±0.19 (+3.65)|89.34±0.41 (+2.88)|
> |CIFAR-10 (α=0.1)|84.42±0.41 (+3.41)|84.94±0.42 (+3.41)|84.11±1.37 (+3.47)|83.63±1.13 (+2.78)|

---

> > ### Comment · Reviewer_djQ3 · 2022-12-04
> > **Reply**
> >
> > > we have assembled a table summarising this information across datasets
> >
> > Thank you, the effect of the fine-tuning stage on the final performance is quite interesting. It seems in the CIFAR dataset, like the bigger the heterogeneity, the bigger impact the fine-tuning stage has on the performance. Could you explain why this is so?
> >
> > > better models are obtained across Tiers showing large gains compared to the original supernet
> >
> > These are impressive results, and it shows that FedorAS can perform well in a bit more realistic setting. However, saying the span of model footprints is "84x" in this experiment is a bit misleading, since the biggest model in Tier 1 (101 MFlops) and the smallest model in Tier 4 (less than 4450M) have less than 44x of a difference. As I understand it, isn't it assumed that all devices participating in Tier 1 has the capacity of performing 101 MFlops computation? If that is the case, 44x is still a small difference compared to the real-world scenario that I have mentioned in the main review.

---

> > > ### Author Response · Authors · 2022-12-05
> > > **Authors' response (2/2)**
> > >
> > > > 44x is still a small difference compared to the real-world scenario that I have mentioned in the main review.
> > >
> > > We appreciate the positive feedback with the new results. In our work we studied whether the exchange of information from different models in terms of their compute footprint could be leveraged. We think it's not misleading to say that the difference between the largest and smallest model in the supernet is 84x (54MFLOPs vs 4450MFLOPs) although we agree it might not be so informative in the context of device capabilities, where the span is indeed 44x. We will make sure the difference is very clear in the revised paper.
> > >
> > > Still, a 44x gap is sufficiently large. To put it into context 44x is more than two times the relative difference between the original tiny [SqueezeNet_1.0](https://pytorch.org/vision/main/models/generated/torchvision.models.squeezenet1_0.html#torchvision.models.squeezenet1_0) and the much larger [ResNeXt101_64x_4d](https://pytorch.org/vision/main/models/generated/torchvision.models.resnext101_64x4d.html). Our largest Tier1 model (101 MFLOPs) is smaller than a vision model that does binary classification on MCUs (best performing model: [VWW-1 MicroNets (see table 4)](https://arxiv.org/pdf/2010.11267.pdf)).
> > >
> > > To further contextualise what a 44x gap is in the context of FL (for the devices that can actually perform these training workloads), we have extracted the following entries from [MLPerf Edge](https://mlcommons.org/en/inference-edge-21/) where the same workload (ResNet-50) is run on three very different devices (that could represent different Tiers in our work): a RaspberryPi4 with a low-end mobile CPU (Cortex A72); a Vim4 development board equipped with mobile-level hardware (Cortex A73+A53, which is equipped in Snapdragon 835: e.g. Google Pixel 2, Samsung S8); and a NVIDIA Jetson Xavier AGX (a high-end embedded GPU system). These devices have the following theoretical peak FP32 FLOPs: 48GFLOPs, 134GFLOPs (vim4 CPU), 307GFLOPs (vim4 Mali GPU), 1410GFLOPs. We can see that our 44x gap could capture well the differences between these devices. Furthermore, in the context of a real workload, despite the large differences in compute/memory capabilities of these platforms (and their price), the relative difference in throughput is less than 10x:
> > >
> > >
> > > | Device | Processor  | Software | Latency (ms) single stream | Throughput |
> > > |:------------- |:-------------:|:-------------:|:-------------:|:-------------:|
> > > |RapsberryPi 4| Arm Cortex-A72| ArmNN v22.05 (Neon) | 332.41 | 3.12|
> > > |RapsberryPi 4| Arm Cortex-A72 | TFLite v2.7.1 (ruy) | 587.70 | 1.74|
> > > |Khadas VIM4 (vim4)| Arm Cortex-A73 + Arm Cortex-A53 | ArmNN v22.05 (Neon) |194.78| 5.18|
> > > |Khadas VIM4 (vim4)| Arm Cortex-A73 + Arm Cortex-A53 + 	ARM Mali G52MP8| ArmNN v22.05 (OpenCL) |149.92	|	6.81|
> > > |Khadas VIM4 (vim4)| Arm Cortex-A73 + Arm Cortex-A53 | TFLite v2.7.1 (ruy) |213.63 | 4.80|
> > > |NVIDIA Jetson AGX Xavier| NVIDIA Carmel (ARMv8.2)| ArmNN v22.05 (Neon) | 72.47	| 17.04
> > > |NVIDIA Jetson AGX Xavier| NVIDIA Carmel (ARMv8.2) | TFLite v2.7.1 (ruy) | 77.98 | 12.99
> > >
> > > Because of the above results (ours and the table above), we argue that the new supernet with 44x FLOPs range is already suitable for many real cross-device FL scenarios with FedorAS being robust and delivering good results in a setting like that. Still, we agree that making this gap even larger (up to the extreme of 500x) would be beneficial and interesting. However, this brings some additional challenges. For example: constructing a supernet that encompasses models of such a different footprint (this hasn't been done before in NAS to the best of our knowledge); or, deal with a much larger memory utilisation during supernet-training (stage 1). Overall, these challenges will require us to re-think our search-space design and/or implementation. To achieve the 500x increase in compute footprint we would likely need to go beyond CIFAR workloads (that use 32x32 inputs) and into ImageNet (224x224 inputs) or different architectures altogether (e.g. transformers), we leave that for future work.

---

> > > ### Author Response · Authors · 2022-12-05
> > > **Authors' response (1/2)**
> > >
> > > > Thank you, the effect of the fine-tuning stage on the final performance is quite interesting. It seems in the CIFAR dataset, like the bigger the heterogeneity, the bigger impact the fine-tuning stage has on the performance. Could you explain why this is so?
> > >
> > >
> > > That's indeed quite interesting - although, we have to keep in mind that even though the improvements from fine-tuning are much higher, the final performance is lower in the absolute sense.
> > > Overall, that suggests a couple of things:
> > >  - supernet training in a highly non-IID setting is more challenging, which follows what is known in conventional FL - this means that the subnets extracted from a supernet in the IID case are already trained close to their full potential, on the other hand non-IID settings are known to converge slower and hence the extracted networks can benefit more from the focused fine-tuning [1,2]
> > >  - orthogonally to the above, we hypothesize that optimising weights in a supernet to work with any architecture not only helps share knowledge between clients, but also acts as a regularizer that mitigates distribution shift that exists in a non-IID setting - let us explain our reasoning behind this hypothesize in more details below:
> > >     - in a conventional FL, different clients optimise $w_i = \texttt{argmin} \mathbb{E}_{(x,y) \sim D_i}L(f(x,w_i), y)$ and we know that as underlying distributions of different $D_i$ overlap less then aggregation becomes more challenging - this is because the specific optimisation objective for different clients (expectation over differently distributed data) pushes $w_i$ into different, less compatiable with each other, directions [2]
> > >     - on the other hand, following the SPOS paper we expect conventional path-sampling NAS to optimise weights of each individual operation (layer) $w_j$ to optimise $\mathbb{E}\_{a \sim A(j)} \mathbb{E}_{(x,y) \sim D} L(f_a(x,w_j), y)$, where $A(j)$ is the set of architectures containing operation $j$ - please note this has only been proposed as an intuitive explanation behind SPOS and no theoretical proof exists yet, consequently our hypothesis also relies on empirical evidence [3]
> > >     -  when we combine both these objectives we get something like: $w\_{i,j} = \texttt{argmin} \mathbb{E}\_{a \sim A(j)} \mathbb{E}\_{(x,y) \sim D_i} L(f_a(x,w_{i,j}), y)$. Importantly, even in the case when datasets residing on different clients $D_i$ are very different, the part of the objective to optimise expected performance over different architectures stays the same and acts as a common factor, preventing "overfitting" of each client's model to its local data, and making subsequent aggregation work better since now weights comming from different clients are trained to optimise more aligned objectives - we further show in our paper that this works especially well if we use our operation aggregation (with improvements ober FedAvg being more visible the more non-IID the setting is);
> > >     - while the above does not - by any means - solve the challenges of training in a non-IID setting completely, our strong empirical results show that the hypothesized regularization is very much helpful
> > >  - because of the above, from a supernet training stage we end up with weights that, intuitively, are optimised to work resonably well regardless of what architecture or client is used for training, resulting in a much more robust initialisation compared to standard FL (rand-init vs. FedorAS), thus the subsequent fine-tuning is able to bring out more potential from the extracted networks
> > >
> > >
> > >
> > > References:
> > >
> > > [1] B. McMahan et al. "Communication-Efficient Learning of Deep Networks from Decentralized Data". AISTATS'17
> > >
> > > [2] X. Li et al. "On the convergence of FedAvg on Non-IDD Data". ICLR'20
> > >
> > > [3] Z. Guo et al. "Single Path One-Shot Neural Architecture Search with Uniform Sampling". ECCV'20

---

### Official Review · Reviewer_7XQj · 2022-11-02

**Confidence:** 5
**Clarity, Quality, Novelty And Reproducibility:** The manuscript is of moderate quality…
**Correctness:** 4
**Technical Novelty And Significance:** 2
**Empirical Novelty And Significance:** 3
**Recommendation:** 6

**Details Of Ethics Concerns:**

N/A.

**Strength And Weaknesses:**

<Strength>

•	The paper is well-organized and well-written.

•	The experimental results supports the claims made in the manuscripts.

•	The experiments and ablation studies are comprehensive.

•	The submitted content is related to the application of neural architecture search in federated learning, which is highly relevant to the ICLR audience.

<Weakness>

•	The novelties of the proposed algorithm are limited. The idea of FedorAS is a direct combination of federated learning and single-path one-shot NAS.

•	Discussion or ablation studies are not sufficient (see below).

<Comments>

1.	What is training and searching efficiency of the proposed algorithm?

2.	How to determine that when to stop the searching process?

3.	Does the training recipe (e.g., batch size) matter for the NAS outcome? Do different training recipes or different random seeds result in different Kendall rank correlation coefficients?

4.	It is the necessary to show the gap between federated NAS (FedorAS) and the centralized NAS?

5.	Is it possible to search for different architectures with FedorAS when clients have different computing devices or budgets?

6.	How does the number of tiers affect the performance of the proposed algorithm?

7.	The necessary ablation study would be evaluating ranking correlations using the proposed super-net based NAS algorithm.

8.	Typo: Page 6, Section 4, “… we we draw …” => “… we draw …”.


**Summary Of The Paper:**

The manuscript proposed a neural architecture search (NAS) algorithm, FedorAS, under cross-device federated learning setting. The proposed algorithm follows settings of one-shot NAS utilizing a super-net search space with weight sharing. It enables knowledge exchange between server and clients via sub-networks. And a new model weight aggregation methods is introduced to compensate sampling frequencies of sub-networks at clients. Moreover, the experimental results validate the proposed algorithm with datasets in computer vision, audio data processing, and natural language processing. The comprehensive experiments cover various FL setting including IID/non-IID data distribution, cross-device FL/cross-silo FL.

**Summary Of The Review:**

The manuscript presented an interesting study of the proposed federated NAS algorithm. Its experiments are comprehensive. But its novelties could be limited as it is a straightforward combination of federated learning and single-path one-shot NAS.

---

> ### Author Response · Authors · 2022-11-17
> **Replying to reviewer's questions (1/2)**
>
> To make this reply self-contained we first indicate the question raised and then add our response.
>
> > _What is training and searching efficiency of the proposed algorithm?_
>
> We have included a new plot (Figure 9) in the appendix illustrating the convergence behaviour of FedorAS through each of its stages. The hyperparameters that control the cost of obtaining these results are summarised in Tab. 7 and 8 in the Appendix. At the same time, we have compiled a table with detailed low-level cost (compute and memory) of running FedorAS (when using typical hyperparameters) and related baselines – please find it in the reply to reviewer 4 (Zb8v).
>
>
> > _How to determine that when to stop the searching process?_
>
> In our experiments, we have terminated the searching process after a set amount of rounds (see "# Search Iterations" in Table 8). Alternatively, one can enforce an accuracy threshold and optionally a number of unique architectures above the threshold, after which search can terminate. For example, in free text this can be similar to "terminate search phase when 10 architectures have over 60% of accuracy on the validation set". We will integrate this discussion in the manuscript.
>
> > _Does the training recipe (e.g., batch size) matter for the NAS outcome? Do different training recipes or different random seeds result in different Kendall rank correlation coefficients?_
>
> Different training recipes do affect the NAS outcome, although this is not more significant than in standard FL or centralised settings. In the context of FedorAS, the combination of batch size and number of local epochs define the exploration vs. exploitation trade-off in the communicated subspace. This happens due to the fact that a different path is sampled for each iteration of a local epoch. It is natural that if not enough paths are explored or if paths overfit the local data, the overall NAS results and thus model ranking would be severely affected.
>
> As explained later - ranking correlation is rather weak and, at the same time, is not of primary interest in the context of our work. Therefore, we did not quantify the effects of random seed on ranking correlation (Figure 4 being the only exception) - however, we do show the impact on the final NAS result through the standard error bars reported throughout the paper.
>
> > _It is the necessary to show the gap between federated NAS (FedorAS) and the centralized NAS?_
>
> While we think this is not directly in scope of our paper's narrative, we provide the requested results to put our method into perspective by showing its performance without the challenges of FL, thus providing an empirical upper-bound for our FL results.
>
> Specifically, to study the impact that FL has in supernet-based NAS, we train the CIFAR-10 supernet in a centralised setting. Similarly to Stage I in FedorAS, we do 500 training epochs following the SPOS paradigm but, because it’s centralised, we do not impose any FLOPs budgets when sampling paths along the supernet. After training, we use NSGA-II to search for the best performing architecture/path and select the best among 1000 valid models for each tier. Finally, these models are trained from scratch for 200 epochs (again in a centralised setting, i.e., having access to all training data). The resulting test accuracies per tier are shown in the table below and compared against the federated IID setting (the simplest FL setup) for FedorAS. We observe a moderate gap in accuracy compared to FL NAS. This is expected, since it is known that underlying nature of the federated process makes the training more challenging (e.g. data partitioning, client sampling,  model aggregation...) .
>
> |  Method | Dataset | Tier 1 | Tier 2 | Tier 3 | Tier 4 |
> |:------------- |:-------------:|:-------------:|:-------------:|:-------------:|:-------------:|
> |Centralised-NAS | CIFAR-10 (centralised) | 93.77±0.20 | 94.14±0.34 | 94.04±0.22 | 94.33±0.09 |
> |FedorAS | CIFAR-10 (α=1000)| 89.40±0.19 (-4.37) | 89.60±0.15 (-4.54) | 89.64±0.22 (-4.40) | 89.24±0.29 (-5.09)|

---

> > ### Author Response · Authors · 2022-11-17
> > **Replying to reviewer's questions (2/2)**
> >
> > > _Is it possible to search for different architectures with FedorAS when clients have different computing devices or budgets?_
> >
> > This is exactly what we had in mind when designing our method. Specifically, we assign different computing budgets (expressed as max FLOPs per forward pass) to different clients which is always honoured, e.g., when sampling a path during stage 1 (Eq. 3 and 4) or during fine-tuning (stage 3) when only eligible clients with sufficient budget are allowed to participate.
> >
> > > _How does the number of tiers affect the performance of the proposed algorithm?_
> >
> > |  Dataset | Method | Tier 1 | Tier 2 | Tier 3 | Tier 4 | Tier 5 | Tier 6 | Tier 7 | Tier 8 |
> > |:------------- |:-------------:|:-------------:|:-------------:|:-------------:|:-------------:|:-------------:|:-------------:|:-------------:|:-------------:|
> > |CIFAR (α=1.0)| supernet  | 87.53±0.29|87.81±0.21|87.59±0.57|87.84±0.26|87.65±0.20|87.76±0.40|87.21±0.55|87.11±0.45|
> > |CIFAR (α=1.0)| _rand-init_ |86.03±0.50|86.74±0.57|85.38±0.43|85.20±0.62|83.98±0.28|83.82±0.65|80.51±0.74|74.26±2.15|
> > |CIFAR (α=0.1)|supernet | 80.34±1.70|81.56±0.46|81.50±1.19|81.54±0.69|81.12±0.81|81.14±0.29|81.28±1.13|79.92±1.94|
> > |CIFAR (α=0.1)| _rand-init_ |68.04±2.22|64.91±3.43|64.84±4.81|65.46±1.98|66.50±2.71|63.50±2.18|63.50±1.09|57.40±1.76|
> >
> > (Please note: the results were obtained by running Stage 1 for 750 epochs, rather than 500, due to an overlooked config parameter. Since we don’t have enough time to rerun the above with 500, please kindly consider these additional results for CIFAR-10$_{\alpha=1.0}$ with 4 tiers and 750 epochs for reference: 88.31(+1.9), 88.17(+2.4), 88.0(+5.6), 87.89(+6.4), where the numbers in parentheses represent improvements over _rand-init_).
> >
> > As we can see, increasing the number of tiers in general makes the setting more challenging. Specifically, this manifests in two ways: 1) results of FedorAS with more tiers are slightly worse compared to results of FedorAS in analogous setting with fewer tiers; but at the same time 2) the gap between FedorAS and naive training of models from the higher tiers is larger - this is a direct consequence of the fact that with more tiers we have fewer clients in each tier resulting in an increased client participation gap (Figure 1) - this follows what we already show in Table 5.
> >
> > > _The necessary ablation study would be evaluating ranking correlations using the proposed super-net based NAS algorithm._
> >
> > Please note that weight-sharing NAS is well-known to fail at producing faithful rankings of architectures in many cases [81,83,85,78,61] - although not formally studied in the federated setting, we’d expect to see a similar trend. Even more, if we consider correlation of scores used by FedorAS with scores obtained from training architectures independently (what is usually considered as ground truth in NAS), it is actually not desired to achieve high correlation due to the client participation gap that we discuss at the end of Section 2 and in Figure 1 (red dots would represent “ground-truth” accuracy in this case). This discrepancy is the main reason why we did not, at first, quantify performance of our method through the lens of ranking correlation - we’d still argue that it has secondary meaning in our setting.
> >
> > Having said that, it is indeed customary to report ranking correlations in NAS papers, so for the sake of completeness please find a summary below - we have also updated our submission to point to a relevant new section in the Appendix.
> >
> > To measure correlations, we randomly sampled 160 architectures from each relevant search space (similar to Figure 1) and subsequently evaluated them in different ways. For CIFAR-10(a=1.0) we include extended evaluation where we consider correlations between any two of the 4 scores used in our paper (Stage 2 acc., fine-tuning acc., standalone acc. (tier-aware) and standalone acc. (tier-unaware)).
> > For CIFAR10(a=0.1) and CIFAR-100(a=0.1) we only include results for Stage 2 acc. vs. fine-tuning acc. as this is the most relevant for our method.
> > The results are visualised in figure 12 in the Appendix (and the ext. CIFAR-10 can bee seen here: https://ibb.co/2nZGVrC).
> >
> > Most importantly, we can see that our searching criteria (val. accuracy after supernet training) correlates rather weakly to the post-fine tuning accuracy (CIFAR-10(a=1.0): 0.255, CIFAR-10(a=0.1): 0.237, CIFAR-100(a=0.1): 0.327).
> > This suggests that finding the best performing architecture is in fact a challenge and that our Stage 2 does not find the optimal solution. However we would argue that this is not necessarily a big issue - specifically, while we cannot easily (i.e., in a cheap way) find the best model, maximising validation accuracy has the upside of avoiding bad models, keeping final accuracy satisfactory (as suggested by our strong results, in addition to the correlation plots shown above). On the other hand, there is clearly room for improvement in this aspect.

---

> ### Author Response · Authors · 2022-12-12
> **Feedback on rebuttal?**
>
> Dear reviewer,
>
> We have greatly appreciated the raised issues and have made reasonable efforts to address the comments in the inital review, wrt:
>
> * searching efficiency
> * stopping criterion
> * ranking correlation of different hyperparameters/recipes
> * scaling evaluation to more tiers of devices
> * federated vs. centralised NAS
> * architectures and different budgets
> * ranking correlations with the proposed NAS technique
>
> It would be greatly appreciated if you could acknowledge whether we addressed your concerns sufficiently and feel more confident about the value of our work.
>
> Please let us know if you have any other thoughts that we can clarify on the paper.
>
> Sincerely,
> The authors

---

### Official Review · Reviewer_C28f · 2022-11-04

**Confidence:** 3
**Correctness:** 3
**Technical Novelty And Significance:** 2
**Empirical Novelty And Significance:** 3
**Recommendation:** 5

**Clarity, Quality, Novelty And Reproducibility:**

Clarity:
(+) The paper elaborates the proposed method in a detailed way, clearly explaining the background, logic of the method and the experiments. The necessary materials are also included in Appendix.
(-) The results are not consistent: In section 4.2, the author claims 48.7% better than CIFAR-10(\alpha=0.1). But in the Tab.2, there is only 26.69 (81.53-54.84) improvement. It is the same for Tab.3. In the content, it says +11.6%, but there is only 9.4 (90.6-81.2). Which is the correct one?
(-) The setting is unclear to me: After supernet training stage, does the system use the same or different model architecture for all clients?
(a) If the model architecture is the same for all clients, how does the proposed method handle the system heterogeneity? That is, how to deal with the clients with different computational resources?
(b) If the model architecture is the different for all clients, is aggregation with OPA still working?

Quality:
(+) This paper provides a detailed analysis of the method, and shows the importance of Supernet initialization and advantage of OPA over FedAvg.

Novelty: The design is based on the idea of supernet, some of the methods are heuristic and may need further analysis or explanation.
(-) (minor) In section 3.1 (1) Subspace sampling, it says “setting the limit to half the size of a typical network … worked well in our experiments …”. Does it apply to other experiments (e.g., different datasets)? Any explanation for this selection?

Reproducibility:
The paper does not provide any code or algorithm block, but it clearly discusses datasets, model search space, and the package it is based on. So probably the results presented in the paper can be reproduced.


**Strength And Weaknesses:**

The paper is well written, and the method is well demonstrated on various datasets compared with different baselines. But there are still some issues, which are discussed in the next part.

**Summary Of The Paper:**

This paper presents FedorAS to discover and train architectures which are suitable for devices of various scales. FedorAS leverages a server-resident supernet enabling weight sharing for knowledge exchange between clients, without sharing common model architectures. This work also provides a new aggregation method called OPA for weighing updates from multiple “single-path one-shot” client updates in a frequency-aware manner. The method is demonstrated on different settings and shows a good performance while maintaining resource efficiency.

**Summary Of The Review:**

The paper is overall well-written, providing a novel approach FedorAS to leverage a supernet for knowledge exchange between clients. The method shows better results by demonstrating on different datasets compared with multiple baselines. However, there are still some fundamental issues that need to be fixed.

---

> ### Author Response · Authors · 2022-11-17
> **Replying to reviewer's questions**
>
> To make this reply self-contained we first indicate the question raised and then add our response.
>
> > _The results are not consistent: In section 4.2, the author claims 48.7% better than CIFAR-10(\alpha=0.1). But in the Tab.2, there is only 26.69 (81.53-54.84) improvement. It is the same for Tab.3. In the content, it says +11.6%, but there is only 9.4 (90.6-81.2). Which is the correct one?_
>
> Throughout our paper, we use X% to indicate relative differences, and X pp or p (percentage points or points respectively) to indicate absolute differences. Following on the example mentioned by the reviewer, 81.53% vs. 54.84% results in an improvement of +26.69pp or +48.7% (26.69/54.84). We will tweak our writing to make sure this difference is clearly conveyed to the readers.
>
> > _The setting is unclear to me: After supernet training stage, does the system use the same or different model architecture for all clients? (a) If the model architecture is the same for all clients, how does the proposed method handle the system heterogeneity? That is, how to deal with the clients with different computational resources? (b) If the model architecture is the different for all clients, is aggregation with OPA still working?_
>
> After supernet has been trained (i.e. stage 1 has finished), our system searches for (stage 2) and fine-tunes (stage 3) a single architecture per device tier. Therefore if we have four tiers, we end up with four architectures which share only part of their initialisation from the supernet before fine-tuning on eligible clients.
> Below we clarify our training process so that questions (a) and (b) are adequately answered.
>
> Both during stage 1 and stage 3, clients are training different models with one another, which are within their budget. In stage 1, they sample from the communicated subspaces, whereas in stage 3 they train the model that belongs to their tier and below (eligibility criterion). As such, FedorAS is always training models on clients in a resource-aware manner. Aggregation with OPA is happening only during stage 1 (see Sec. 3.1/point 3), where different parts of the supernet have been communicated and different paths have been trained on clients based on their tier dynamics. During stage 3, aggregation happens with conventional FedAvg across participating clients (see Sec. 3.3) because they are finetuning a standard model (i.e. no longer a sub-supernet).
>
>
> > _(minor) In section 3.1 (1) Subspace sampling, it says “setting the limit to half the size of a typical network … worked well in our experiments …”. Does it apply to other experiments (e.g., different datasets)? Any explanation for this selection?_
>
> Yes, since empirical results were satisfactory for CIFAR-10, we adopted similar setups for all datasets/models. We provide extra insights about this very behaviour in Section 4.4 and Appendix G.1 of our manuscript. Specifically, we conducted detailed experiments about the impact of the communicated subspaces and aggregation method on CIFAR-10$_{α=1.0}$. Figure 8 of the Appendix shows the convergence behaviour and performance per tier.

---

> ### Author Response · Authors · 2022-12-12
> **Feedback on rebuttal?**
>
> Dear reviewer,
>
> We value the effort put into reviewing our manuscript and your comments.
>
> We have addressed all the raised concerns to the best of our ability, along with providing the code for our system.
> We would greatly appreciate it if you could let us know whether we have covered your concerns sufficiently and you feel more confident about the value of our work.
>
> Please let us know if you have any other thoughts that we can clarify on the paper.
>
>
> Sincerely,
> The authors

---

### Author Response · Authors · 2022-11-17
**Rebuttal summary**

We thank the reviewers for their feedback on our work and the points raised that need further clarification. We have addressed all questions raised by reviewers individually. When a point was considered by more than one reviewer we still wrote a reply to each but often referred one answer to one we gave for the other reviewer. The feedback given by reviewers as well as the additional experiments conducted in this rebuttal have been incorporated to the paper (either to the main text or to the appendix) and are shown in blue font.

**Summary of new content:** comparison of FL-NAS against centralised NAS; results with doubled the number of tiers; ablations of the improvement obtained during finetuning; results for supernet with much wider range of FLOPs (tier 4 models are up to 82x large than tier 1 models); plots showing end-to-end convergence behaviour of FedorAS for all datasets; comprehensive analysis of compute and memory footprint of Fedoras and baselines; ranking correlation analysis; detailed comparison against HeteroFL.

**Common questions:**
- We were asked if FedorAS supports more tiers than the ones evaluated in our manuscript. Yes, it does. Results are reported for reviewer 7XQj.
- We were asked if the code would be open-sourced: yes, it will be open sourced. In the meantime, find it following this https://github.com/anonymous-fedora/FedorAS23  (we’ll share the password to the zip file in a separate comment available only to reviewers). We do it in this way because we are still going through the legal process required by our team for open-sourcing code.
- We were asked about the costs of obtaining the final models and the overall efficiency of FedorAS: a detailed analysis has been given to reviewer Zb8v and a new figure has been added to the Appendix (Figure 9): overall, FedorAS is more memory efficient than the competing FL+NAS framework even when using larger batch size; compared to the competing tier-aware method (FjORD), FedorAS largely outperforms it while keeping compute and memory footprints under control taking into account that FedorAS delivers a uniquely customised architecture for each tier and tasks (i.e. NAS).

---

### Author Response · Authors · 2022-11-24
**Looking forward to post-rebuttal feedback**

Dear reviewers,

To the best of our ability, during rebuttal we have address the points raised and revised the submitted manuscript (updates in blue) incorporating your feedback. We would like to hear your thoughts and revised scores if applicable.

Kind regards,
The authors

---

### Decision · Program_Chairs · 2023-01-20

**Decision:**

Reject

**Justification For Why Not Higher Score:**

The proposed method is not fully validated for its efficiency, which is claimed to be its main strength.

**Justification For Why Not Lower Score:**

N/A

**Metareview: Summary, Strengths And Weaknesses:**

This paper proposes a federated NAS framework targeting cross-device scenarios, which focuses on the device heterogeneity and communication efficiency. Specifically, the proposed FedorAS framework trains the supernet via federated learning across multiple devices, where each device performs budget-aware path sampling and trains the subnetwork on the local data, which are aggregated by considering the number of updates to each operation per client. The proposed FedorAS framework is validated on multiple benchmarks with both IID and non-IID partitioning, and is shown to largely outperform the compared baselines (e.g. FedNAS, FjORD).

The paper initially received split reviews, with two leaning toward acceptance and two others leaning toward rejection. The reviewers found the paper well-written, the method clear, the experimental validation extensive, and the performance of the proposed method impressive. However, the reviewers were also concerned about the following factors:
- Limited novelty of FedorAS over existing federated NAS frameworks, such as FedNAS, or heterogenous FL frameworks, such as HeteroFL, since neither the idea of learning NAS frameworks in an FL framework, or federated learning of a supernet, is novel. The main novelty thus reduces down to the proposal of an efficient and device-aware federated NAS method.
- There is hidden cost in the supernet training as well as the finetuning, that is not reported in the submitted version of the paper. Also, the reduction in the communication cost by sending 25% or 50% of the supernet may not be that large if the original network is large.
- The "tiers" and the number of tiers are manually defined and the experimental setting does not consider the scenario where there exists large difference in the computing power among the devices.

During the discussion phase, the authors provided detailed responses to the reviewers with additional experimental results. However, the paper still fell into the borderline case even after the rebuttal, and thus the reviewers had in-depth discussion over zoom. From the zoom meeting, the reviewers agreed that the paper presents promising results and the method thus may have some practical impact. However, they considered the consideration of cross-device setting as incremental over existing fedNAS frameworks, and most importantly, they were concerned with the hidden cost and insufficient reduction in the communication cost. These are critical weaknesses since they may go against the main argument that the proposed method is more efficient than the baselines, in terms of the overall training and communication cost. Thus, the main strength of the model now comes from its good performance, but the reviewers agreed that there is little analysis of how the proposed method could obtain such impressive performance.

In sum, while the paper proposes a potentially promising framework for cross-device federated learning problems, its main argument that the proposed method is more efficient is not properly validated, and the effectiveness of the proposed method is not fully analyzed. Thus, the paper could be accepted if there is a room, but will clearly benefit from another round of revision, if that is not the case.

**Summary Of Ac-Reviewer Meeting:**

From the zoom meeting, the reviewers agreed that the paper presents promising results and the method thus may have some practical impact. However, they considered the consideration of cross-device setting as incremental over existing fedNAS frameworks, and most importantly, they were concerned with the hidden cost and insufficient reduction in the communication cost. These are critical weaknesses since they may go against the main argument that the proposed method is more efficient than the baselines, in terms of the overall training and communication cost. The reviewers were also concerned on the lack of analysis on how the proposed method could obtain such impressive performance over existing fedNAS frameworks.

Overall, the reviewers were not overly negative and found the work promising, but they unanimously agreed that the paper are not ready for publication yet due to insufficient validation and analysis.